# Efficient probabilistic inference in generic neural networks trained with non-probabilistic feedback

A. Emin Orhan[1] & Wei Ji Ma[1,2]

Animals perform near-optimal probabilistic inference in a wide range of psychophysical tasks. Probabilistic inference requires trial-to-trial representation of the uncertainties associated with task variables and subsequent use of this representation. Previous work has implemented such computations using neural networks with hand-crafted and task-dependent operations. We show that generic neural networks trained with a simple error-based learning rule perform near-optimal probabilistic inference in nine common psychophysical tasks. In a probabilistic categorization task, error-based learning in a generic network simultaneously explains a monkey's learning curve and the evolution of qualitative aspects of its choice behavior. In all tasks, the number of neurons required for a given level of performance grows sublinearly with the input population size, a substantial improvement on previous implementations of probabilistic inference. The trained networks develop a novel sparsity-based probabilistic population code. Our results suggest that probabilistic inference emerges naturally in generic neural networks trained with error-based learning rules.

[1] Center for Neural Science, New York University, New York, NY 10003, USA. [2] Department of Psychology, New York University, New York, NY 10003, USA. Correspondence and requests for materials should be addressed to A.E.O. (email: aeminorhan@gmail.com) or to W.J.M. (email: weijima@nyu.edu)

When faced with noisy and incomplete sensory information, humans and other animals often behave near-optimally[1–6]. In many tasks, optimal behavior requires that the brain compute posterior distributions over task-relevant variables, which often involves complex operations such as multiplying probability distributions or marginalizing over latent variables. How do neural circuits implement such operations? A prominent framework addressing this question is the probabilistic population coding (PPC) framework, according to which the population activity on a single trial encodes a probability distribution rather than a single estimate and computations with probability distributions can be carried out by suitable operations on the corresponding neural responses[7, 8]. For example, Ma et al.[7] showed that if neural variability belongs to a particular class of probability distributions, the posterior distribution in cue combination tasks can be computed with a linear combination of the input responses. Moreover, in this scheme, the form of neural variability is preserved between the input and the output, leading to an elegantly modular code. In more complex tasks, linear operations are insufficient and it has been argued that multiplication and division of neural responses are necessary for optimal inference[9–13].

Upon closer look, however, these previous implementations of PPC suffer from several shortcomings. First, the networks in these studies were either fully manually designed, or partially manually designed and partially trained with large amounts of probabilistic data to optimize explicitly probabilistic objectives, for example, minimization of Kullback-Leibler (KL) divergence. Therefore, this literature does not address the important question of learning: how can probabilistic inference be learned from a realistic amount and type of data with minimal manual design of the networks? Second, although there are some commonalities

between the neural operations required to implement probabilistic inference in different tasks, these operations generally differ from task to task. For instance, it has been argued that some form of divisive normalization of neural responses is necessary in tasks that involve marginalization[10]. However, the specific form of divisive normalization that individual neurons have to perform differs substantially from task to task. Therefore, it is unclear if probabilistic inference can be implemented in generic neural networks, whose neurons all perform the same type of neurally plausible operation. Third, in these studies, the number of neurons used for performing probabilistic inference scales unfavorably with the size of the input population (linearly in the case of cue combination, but at least quadratically in all other tasks). Therefore, the question of whether these tasks can be implemented more efficiently remains open.

In this paper, we address these issues. We show that generic neural networks trained with non-probabilistic error-based feedback perform near-optimal probabilistic inference in tasks with both categorical and continuous outputs. Generic neural networks of the type we use in this paper have a long history[14–16], and have recently been linked directly to cortical responses[17–20]. Our main contribution is to connect generic neural networks to near-optimal probabilistic inference in common psychophysical tasks. For these tasks, we analyze the network generalization performance, the efficiency of the networks in terms of the number of neurons needed to achieve a given level of performance, the nature of the emergent probabilistic population code, and the mechanistic insights that can be gleaned from the trained networks. We also investigate whether the time course of error-based learning in generic neural networks is realistic for a non-linguistic animal learning to perform a probabilistic inference task.

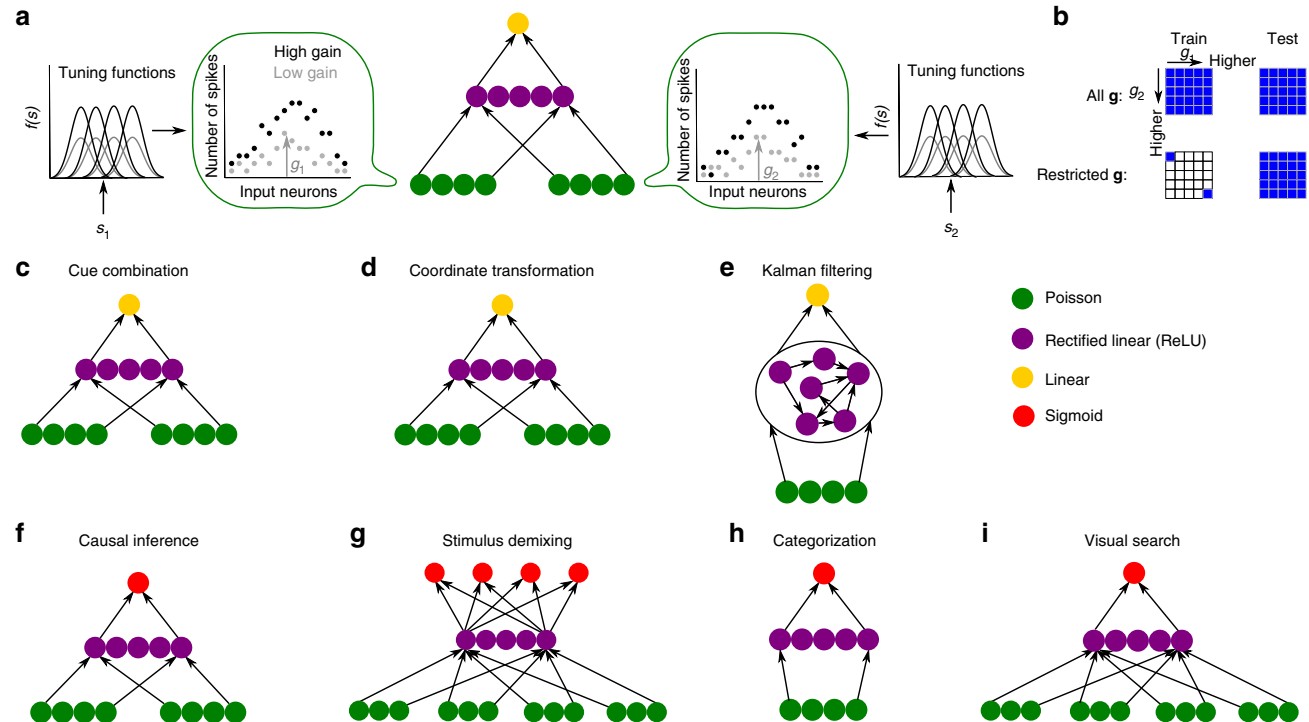

**Fig. 1** General task set-up and network architectures. **a** General task set-up. Input populations encode possibly distinct stimuli, $s_i$, with Poisson noise. The amount of noise was controlled through multiplicative gain variables, $g_i$, which varied from trial to trial. **b** Training and test conditions. In the "all **g**" condition, the networks were trained on all possible gain combinations (represented by the *blue tiles*); whereas in the "restricted **g**" condition, they were trained on a small subset of all possible gain combinations. In both conditions, the networks were then tested on all possible gain combinations. **c–i** Network architectures used for the seven main tasks. Different *colors* represent different types of units. For tasks with continuous output variables **c–e**, linear output units were used; whereas for tasks with categorical output variables **f–i**, sigmoidal output units were used

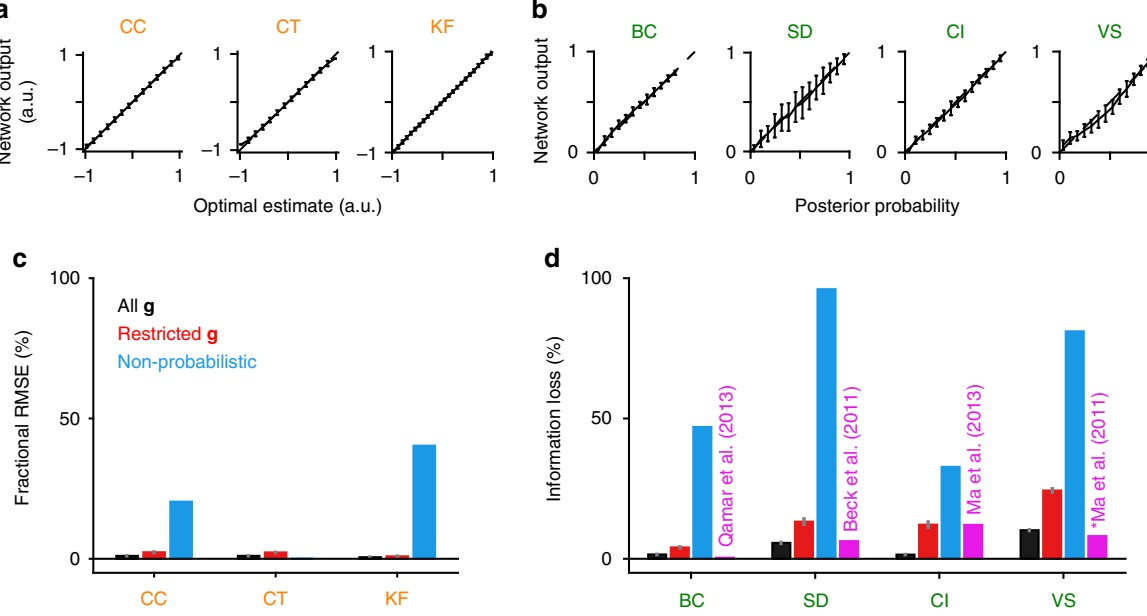

**Fig. 2** Performance of well-trained networks in the main tasks. **a** Optimal estimates vs. the network outputs in "all **g**" conditions of the continuous tasks. *Error bars* represent standard deviations over trials. **b** Posterior probability of a given choice vs. the network output in "all **g**" conditions of the categorical tasks. *Error bars* represent standard deviations over trials. For the SD task, where the output was four-dimensional, only the marginal posterior along the first dimension is shown for clarity. **c** Performance in continuous tasks. **d** Performance in categorical tasks. *Blue bars* show the performance of non-probabilistic heuristic models that do not take uncertainty into account. Note that optimal performance in the CT task does not require taking uncertainty into account (see "Methods" section). *Magenta bars* show the performance of hand-crafted networks in categorical tasks reported in earlier works. The *asterisk* in the VS task indicates that the information loss value reported in ref. [11] should be taken as a lower bound on the actual information loss, since they were not able to build a single network that solved the full visual search task in that paper. In **c**, **d**, error bars (*gray*) represent means and standard errors over 10 independent runs of the simulations. *CC* cue combination, *CT* coordinate transformation, *KF* Kalman filtering, *BC* binary categorization, *SD* stimulus demixing, *CI* causal inference, *VS* visual search. Categorical tasks are labeled in *green*, continuous tasks in *orange*

## Results

**Tasks**. We trained generic feedforward or recurrent neural networks on nine probabilistic psychophysical tasks that are commonly studied in the experimental and computational literature. The main tasks were linear cue combination[1–3], coordinate transformation[10, 16], Kalman filtering[5, 6, 21], causal inference[4, 12], stimulus demixing[10], binary categorization[13], and visual search with heterogeneous distractors[11] (see "Methods" section for task details). We also trained generic networks on two additional "modular" tasks to be discussed below.

**Networks**. The networks all received noisy sensory information about the stimulus or the stimuli in the form of a neural population with Poisson variability (Fig. 1a). The hidden units of the networks were modeled as rectified linear units (ReLUs). ReLUs are commonly used in neural networks due to their demonstrated advantage over alternative non-linearities in gradient-based learning algorithms[22]. Linear (sigmoidal) output units were used in tasks with continuous (categorical) outputs. Schematic diagrams of the networks used for the seven main tasks are shown in Fig. 1c–i. The Kalman filtering task requires memory, and is thus implemented with a generic recurrent network. Other differences between the network architectures are due entirely to differences in the input and output requirements of different tasks: different tasks have different numbers of inputs or outputs and the outputs are continuous or categorical in different tasks. Other than these task-dictated differences, the networks are generic in the sense that they are composed of neurons that perform the same type of biologically plausible operations in all tasks.

Networks were trained to minimize mean squared error or cross-entropy in tasks with continuous or categorical outputs,

respectively. Importantly, the networks were provided only with the actual stimulus values or the correct class labels as feedback in each trial. Thus, they did not receive any explicitly probabilistic feedback, nor were they explicitly trained to perform probabilistic inference.

For the main experiments, we manipulated sensory reliability trial by trial via gain variables **g** multiplying the mean responses of the input populations, with higher gains corresponding to more reliable sensory information (Fig. 1a). We later consider alternative ways of manipulating the sensory reliability (see "Alternative representations of sensory reliability" section below). In each task, networks were tested with a wide range of gains or gain combinations (in tasks with more than a single stimulus). To test the generalization capacity of the networks, we trained them with a limited range of gains or gain combinations, as well as with the full range of test gains or gain combinations. The latter unrestricted training regime is called the "all **g**" condition, whereas the former limited training regime is called the "restricted **g**" condition in what follows (Fig. 1b). The specific gain ranges and gain combinations used in each task are indicated in the "Methods" section.

**Trained generic networks implement probabilistic inference**. The performance of well-trained networks is shown in Fig. 2c, d for both "all **g**" (black) and "restricted **g**" (red) training conditions in all tasks (learning curves of the networks are shown in Supplementary Fig. 1). For continuous tasks, performance is measured in terms of fractional RMSE defined as $100 \times (\text{RMSE}_{\text{netw}} - \text{RMSE}_{\text{opt}})/\text{RMSE}_{\text{opt}}$, where $\text{RMSE}_{\text{netw}}$ is the root mean squared error (RMSE) of the trained network, $\text{RMSE}_{\text{opt}}$ is the RMSE of the posterior mean estimate. For categorical tasks, performance is measured in terms of

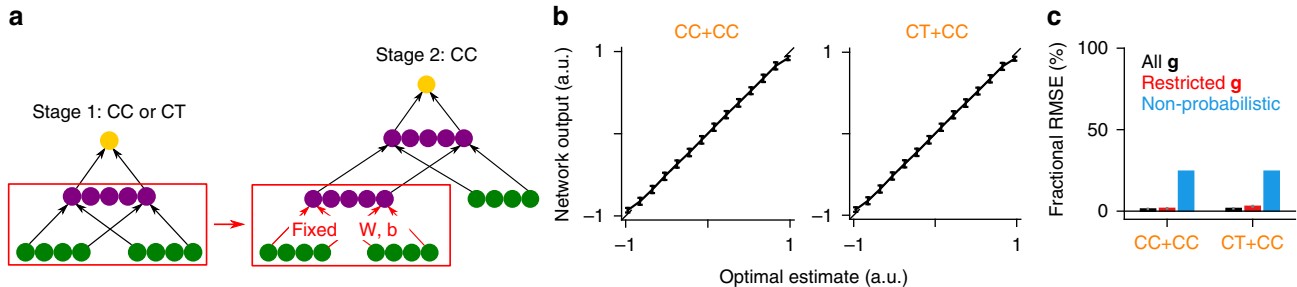

**Fig. 3** Modular tasks for testing the encoding of posterior uncertainty. **a** Schematic diagram illustrating the design of the modular tasks. In stage 1, a single hidden layer network is trained on either cue combination (CC) or coordinate transformation (CT). In stage 2, the hidden layer of the trained network is plugged into another network. The combined network is trained on a three-input CC task with the parameters of the first network fixed. **b** Optimal estimates vs. network outputs for the three-input CC task. *Error bars* represent standard deviations over trials. CT + CC is the case where the first network is trained on the coordinate transformation task, CC + CC is the case where the first network is trained on the two-input CC task. **c** Fractional RMSEs. *Error bars* represent standard errors over 10 independent runs of the simulations. The combined networks perform the three-input CC task near optimally, suggesting that the initial networks encode posterior uncertainty for the first two inputs

fractional information loss defined as the average KL-divergence between the actual posterior and the network's output normalized by the mutual information between the class labels and the neural responses[13]. With this measure, a network that exactly reproduces the posterior achieves 0% information loss, whereas a network that produces random responses according to the prior probabilities of the classes has 100% information loss. Figure 2a, b shows the optimal outputs vs. the network outputs in "all **g**" conditions of the continuous and categorical tasks, respectively (see Supplementary Figs. 2–3 for the optimal vs. network estimates for individual gain combinations in the cue combination and coordinate transformation tasks).

To make sure that optimal performance in our tasks cannot be easily mimicked by heuristic, non-probabilistic models, we also calculated the performance of non-probabilistic reference models that did not take the reliabilities of the inputs into account (Fig. 2c, d, *blue*). In continuous tasks, the non-probabilistic models estimated the individual cues or inputs optimally, but combined them suboptimally by weighing them equally regardless of their reliability. Note that these models still performed a non-trivial probabilistic computation, namely marginalizing out a nuisance variable, i.e., the input gain, to come up with the optimal estimate of the individual cues. Similarly, in categorical tasks, the non-probabilistic models replaced the different reliability terms for different inputs in the optimal decision rules by a common reliability term (see "Methods" section). The large performance gaps between these non-probabilistic models and the optimal model suggest that approaching optimal performance in our tasks requires correctly taking the reliabilities of the inputs into account.

In categorical tasks, the output nodes of the networks approximate the posterior probabilities of the classes given the inputs (Fig. 2b, d). Theoretical guarantees ensure that this property holds under general conditions with a wide range of loss functions[23] (see "Discussion" section).

**Encoding of posterior width in the hidden layers**. In continuous tasks, training with the mean squared error loss guarantees asymptotic convergence to the posterior mean. Do the networks also represent information about the posterior uncertainty in their hidden layers or do they discard this information? Representation of posterior uncertainty is evident in the Kalman filtering task, where accurate encoding of the posterior mean at a particular moment already requires the encoding of the posterior mean and the posterior width at the previous moment and the

optimal integration of these with the current sensory information in the recurrent activity of the network.

For the linear cue combination and coordinate transformation tasks, to test for the representation of posterior uncertainty in the hidden layer, we plugged the trained hidden layers into a network incorporating an additional input population and fixed their parameters (Fig. 3a). The rest of the network was then trained on a linear cue combination task with three input populations (a similar modular task was designed in ref. [24]). If the fixed hidden layers do not encode the posterior width for the first two inputs, the combined network cannot perform the three-input cue combination task optimally. However, the combined networks were able to perform the three-input cue combination task with little information loss despite receiving information about the first two inputs only through the fixed hidden layers (Fig. 3b, c). This suggests that although the initial networks were trained to minimize mean squared error, and hence were asymptotically guaranteed to reproduce the posterior means only, information about the posterior widths was, to a large extent, preserved in the hidden layer. The precise format in which posterior uncertainty is represented in the hidden layer activity will be discussed in detail later (see "Sparsity-based representation of posterior uncertainty" section below). The combined coordinate transformation-cue combination (CT + CC) network also illustrates the generic nature of the representations learned by the hidden layers of our networks: the hidden layer of a network trained on the coordinate transformation task can be combined, without modification, with an additional input population to perform a different task, i.e., cue combination in this example.

**Generalization to untrained stimulus conditions**. It has been argued that truly Bayesian computation requires that the components of the Bayesian computation, i.e., sensory likelihoods and the prior, be individually meaningful to the brain[25]. Thus, if we replace a particular likelihood for another, the system should continue to perform near-optimally. We tested for a limited form of such "Bayesian transfer" by examining whether the trained networks generalize to unseen values or combinations of sensory reliability ("restricted **g**" conditions). As shown in Fig. 2c, d (*red bars*), the networks were able to generalize well beyond the training conditions in all tasks. An example is shown in Fig. 4a for the cue combination task. In this example, we trained a network with only two gain combinations, $\mathbf{g} \equiv (g_1, g_2) = (5, 5)$ and $\mathbf{g} = (25, 25)$, and tested it on all gain combinations of the form $(g_1, g_2)$, where $g_1, g_2 \in \{5, 10, 15, 20, 25\}$ with up to fivefold gain differences between the two input populations (note that

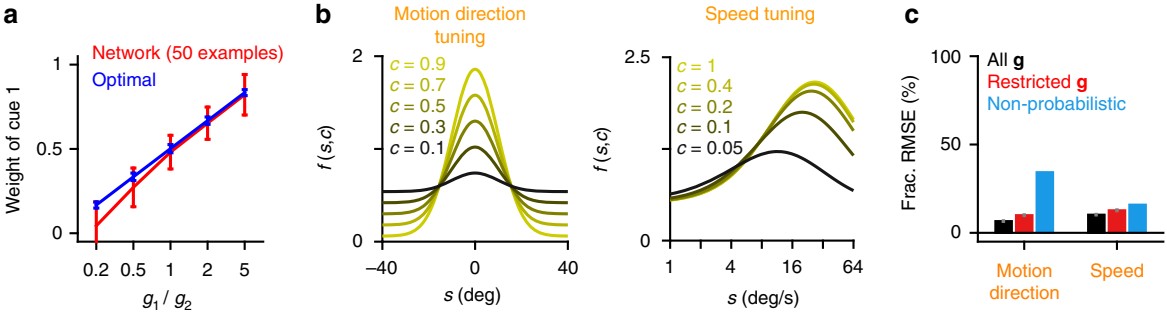

**Fig. 4** Alternative representations of sensory reliability and generalization to unseen gain conditions. **a** Generalization capacity of neural networks in the cue combination task. The weights assigned to cue 1 in the cue conflict trials as a function of the ratio of the input gains, $g_1/g_2$. In cue conflict trials, $s_1$ was first drawn uniformly from an interval of length $l$, then $s_2$ was generated as $s_1 + \Delta$ where $\Delta$ was one of $-2l$, $-3l/2$, $-l$, $-l/2$, $l/2$, $l$, $3l/2$, and $2l$. The cue weight assigned by the network was calculated through the equation: $\hat{s} = w\hat{s}_{1,\mathrm{opt}} + (1 - w)\hat{s}_{2,\mathrm{opt}}$, where $\hat{s}$ is the network output, $\hat{s}_{1,\mathrm{opt}}$ and $\hat{s}_{2,\mathrm{opt}}$ are the optimal estimates of $s_1$ and $s_2$. Note that we use relatively high gains in this particular example to make the optimal combination rule approximately linear. In the low-gain conditions used in the main simulations, the optimal combination rule is no longer linear. The network experienced only 50 training examples in the "restricted **g**" condition with non-conflicting cues, hence it did not see any of the conditions shown here during training. *Error bars* represent standard deviations over 1000 simulated trials. **b** Tuning functions for motion direction as reported in refs. [28, 29] and for speed as reported in ref. [30], where the stimulus contrast or coherence variable $c$ does not act multiplicatively on the mean firing rates. **c** Fractional RMSEs in CC tasks with tuning functions shown in **b**. The networks perform near-optimal probabilistic inference in both cases demonstrating the robustness of our approach to variations in the specific form in which stimulus reliability is encoded in the input populations

these gains are higher than those used in the main simulations to make the optimal combination rule approximately linear). To demonstrate that the trained networks performed qualitatively correct probabilistic inference, we set up cue conflict conditions similar to the cue conflict conditions in psychophysical studies[2], where we presented slightly different stimuli to the two input populations and manipulated the degree of conflict between the cues. The weights assigned to the first cue as a function of the gain ratio, $g_1/g_2$, are shown in Fig. 4a both for the network and for the optimal rule. The network achieved low generalization error (fractional RMSE: 10.9%) even after as few as 50 training examples in the impoverished training condition and performed qualitatively correct probabilistic inference in the untrained conditions. In particular, the network correctly adjusted the weights assigned to the two cues even as the ratio of their reliabilities varied over a 25-fold range (Fig. 4a).

The successful generalization performance of the neural networks is a result of two factors. First, the target function is invariant, or approximately invariant, to some of the gain manipulations that differ between the training and test conditions. In cue combination, for instance, the target function is invariant to the scaling of the input populations by a common gain $g$ (Eq. 5). The second factor is the network's inductive biases, i.e., how it tends to behave outside the training domain. These inductive biases depend on the details of the network architecture[26].

**Alternative representations of sensory reliability**. Thus far, we have assumed that sensory reliability has a purely multiplicative effect on the responses of input neurons. Although this assumption likely holds for the effect of contrast on orientation selectivity in visual cortex[27], it is known to be violated for the effect of motion coherence on direction selectivity[28, 29] and for the effect of contrast on speed selectivity[30], and is unlikely to hold in the general case. The importance of this observation is that the linear Poisson-like PPC approach proposed in ref. [7] cannot handle cases where "nuisance variables" such as contrast or coherence do not have a purely multiplicative effect on neural responses. By contrast, our approach does not make any restrictive assumptions about the representation of stimulus reliability in the input populations. We demonstrated this in two

cases (Fig. 4b, c): (i) cue combination with tuning functions of the form reported in refs. [28, 29], where stimulus coherence affects both the gain and the baseline of the responses (Fig. 4b, *left*) and (ii) cue combination with tuning functions of the form reported in ref. [30] for speed, where both the peak response and the preferred speed depend on stimulus contrast (Fig. 4b, *right*). These results provide evidence for the robustness of our approach to variations in the way in which sensory reliability is encoded in the input populations.

**Sparsity-based representation of posterior uncertainty**. We now discuss how posterior uncertainty is represented in the hidden layers of the trained networks. This discussion applies only to our main experiments where sensory reliability in the input populations is manipulated through purely multiplicative gain. We first note that in tasks with continuous output variables, the optimal solution is invariant to a multiplicative scaling $g$ of the input responses (see "Methods" section, Eqs. (5)–(8)). In such gain-invariant (or approximately gain-invariant) tasks, we find that posterior uncertainty is represented in the sparsity of hidden layer activity. To understand the mechanism through which this sparsity-based representation arises, we investigated the conditions under which the network's output would be invariant to input gain scalings. We first derived an approximate analytical expression for the mean response of a typical hidden unit $\overline{\mu}$, as a function of the input gain $g$, the mean input $\mu$ to the hidden unit for unit gain, and the mean $\mu_b$ and the standard deviation $\sigma_b$ of the biases of the hidden units (see "Methods" section). To minimize the dependence of the mean hidden unit response on $g$, we introduced the following measure of the total sensitivity of $\overline{\mu}$ to variations in $g$:

$$T_{\mathrm{var}} = \int_{g_{\min}}^{g_{\max}} |\overline{\mu}'(g)| \, \mathrm{d}g$$

where the prime represents the derivative with respect to $g$, and numerically minimized $T_{\mathrm{var}}$ with respect to $\mu$, $\mu_b$, and $\sigma_b$, subject to the constraint that the mean response across different gains be equal to a positive constant $K$. $T_{\mathrm{var}}$ was minimized for a negative mean input $\mu$, positive $\mu_b$, and a large $\sigma_b$ value (*black star* in Fig. 5a). We note that because the input responses are always non-negative, the only way $\mu$ can be negative in our networks is if

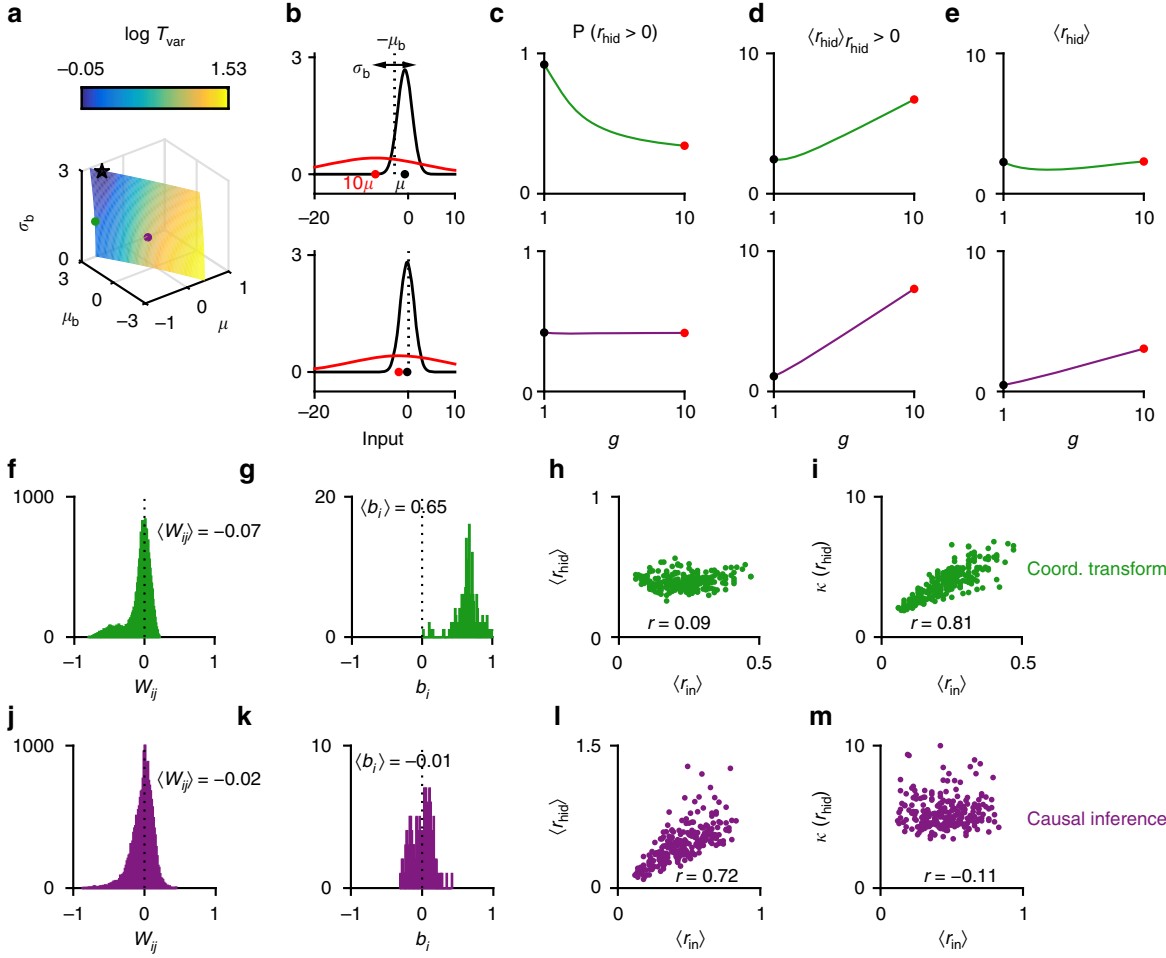

**Fig. 5** The mechanism underlying the sparsity-based representation of posterior uncertainty. **a** The variability index $T_{var}$ (plotted in log units) as a function of the parameters $\mu$, $\mu_b$, and $\sigma_b$ for the constraint surface corresponding to $K = 2$. The optimal parameter values within the shown range are represented by the *black star*. The *green* and *magenta dots* roughly correspond to the parameter statistics in the trained coordinate transformation and causal inference networks, respectively. **b** For the parameter combination corresponding to the *green dot*, the mean input $\mu$ is negative and the mean bias $\mu_b$ is positive. Increasing the gain $g$ thus shifts the input distribution to the *left* and widens it: compare the *black* and *red lines* for input distributions with a small and a large gain, respectively. The means of the input distributions are indicated by *small dots* underneath. **c** This, in turn, decreases the probability of non-zero responses, but **d** increases the mean of the non-zero responses; hence, **e** the mean response of the hidden units, being a product of the two, stays approximately constant as the gain is varied. In the *bottom panel* of **b**–**e**, the results are also shown for a different parameter combination represented by the *magenta dot* in **a**. This parameter combination roughly characterizes the trained networks in the causal inference task. **f** For a network trained in the coordinate transformation task, distributions of the input-to-hidden layer weights, $W_{ij}$; **g** the biases of the hidden units, $b_i$; **h** scatter plot of the mean input $\langle r_{in} \rangle$ vs. the mean hidden unit activity $\langle r_{hid} \rangle$; **i** the scatter plot of the mean input vs. the kurtosis of hidden unit activity $\kappa(r_{hid})$. **j**–**m** Similar to **f**–**i**, but for a network trained in the causal inference task

the mean input-to-hidden layer weight is negative. As an approximate rule, decreasing $\mu$ and increasing $\mu_b$ or $\sigma_b$ lead to smaller $T_{var}$ values. A large $\mu_b$ causes a large proportion of the input distribution to be above the threshold for low gains. The negativity of the mean input $\mu$ implies that as the gain $g$ increases, the distribution of the total input to the unit shifts to the left (Fig. 5b, *top*) and becomes wider, causing a smaller proportion of the distribution to remain above the threshold (represented by the *dashed line* in Fig. 5b), hence decreasing the probability that the neuron will have a non-zero response (Fig. 5c, *top*). This combination of large positive $\mu_b$ and negative $\mu$ causes the sparsification of the hidden unit responses with increasing $g$. Because increasing $g$ also increases the variance of the total input to the unit, the mean response for those inputs that do cross the threshold increases (Fig. 5d, *top*). As a result, the mean response of the neuron, which is a product of these two terms, remains roughly constant (Fig. 5e, *top*).

We demonstrate this sparsification mechanism for a network trained on the coordinate transformation task in Fig. 5f–i. Because the coordinate transformation task is approximately gain-invariant (see "Methods" section, Eq. (6)), the input-to-hidden layer weight distribution in the trained network was skewed toward negative values (Fig. 5f) and the mean bias of the hidden units, $\mu_b$, was positive (Fig. 5g), as predicted from our simple mean-field model. Consequently, we found a strong positive correlation between the sparsity of hidden layer responses and the mean input response ($r = 0.81$, $P < 10^{-6}$; Fig. 5i), but no correlation between the mean hidden layer response and the mean input response ($r = 0.09$, $P > 0.05$; Fig. 5h).

The same type of analysis applies to the categorical tasks as well. However, the difference is that for some of our tasks with categorical outputs, in the optimal solution, the net input to the output unit had a strong dependence on $g$.

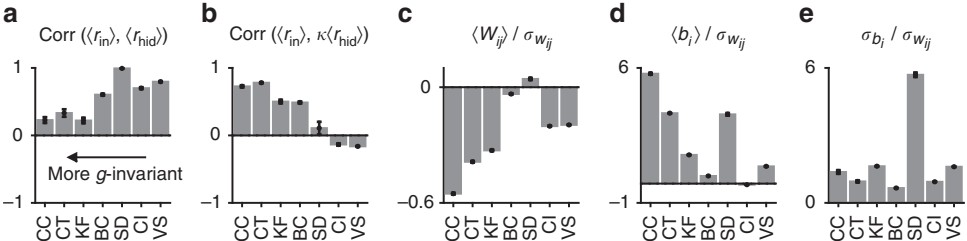

**Fig. 6** Encoding of posterior uncertainty and parameter statistics in the trained networks. **a** Trial-by-trial correlation between mean hidden unit response and mean input response; **b** trial-by-trial correlation between the sparsity (kurtosis) of hidden layer activity and mean input response; **c** mean input-to-hidden unit weight; **d** mean bias of the hidden units; **e** standard deviation of hidden unit biases. Parameter statistics are reported in units of the standard deviation of the input-to-hidden layer weights, $\sigma_{W_{ij}}$, to make them consistent with the mean field analysis, where all variables are measured in units of the standard deviation of the input. *CC* cue combination, *CT* coordinate transformation, *KF* Kalman filtering, *BC* binary categorization, *SD* stimulus demixing, *CI* causal inference, *VS* visual search. *Error bars* represent means and standard errors over 10 independent runs of the simulations

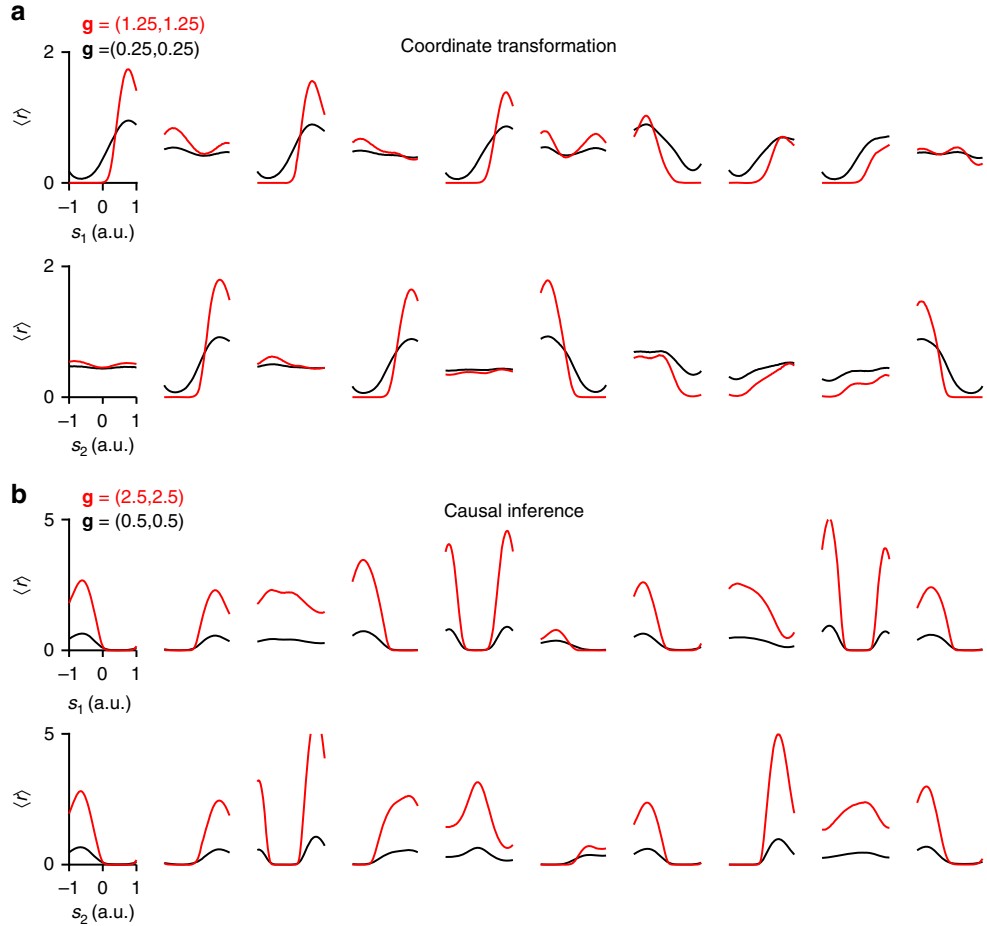

**Fig. 7** One-dimensional tuning functions of 10 representative hidden units at two input gains **a** in a network trained in the coordinate transformation task and **b** in a network trained in the causal inference task. In both **a** and **b**, the *first row* shows the tuning functions with respect to $s_1$ (averaged over $s_2$) and the *second row* shows the tuning functions with respect to $s_2$. Increasing the gain sharpens the tuning curves in the coordinate transformation task, whereas it has a more multiplicative effect in the causal inference task

For example, in causal inference, the input to the sigmoidal output unit scales approximately linearly with $g$ (Eq. 10). Similarly, in visual search, both global and local log-likelihood ratios have a strong dependence on $g$ (through $\mathbf{r}_i$ in Eqs. (16) and (17)). We emphasize that the distinction between $g$-dependence and $g$-invariance is not categorical: different tasks can have varying degrees of $g$-invariance or $g$-dependence and parameter choices in the same task can affect its $g$-dependence.

In the *bottom panel* of Fig. 5b–e, predictions from the mean-field model are shown for a parameter combination where both $\mu$ and $\mu_b$ are small and slightly negative (represented by the *magenta dot* in Fig. 5a). This parameter combination roughly characterizes the trained networks in the causal inference task (Fig. 5j–m). In this case, because both $\mu$ and $\mu_b$ are close to 0, the probability of non-zero responses as a function of $g$ stays roughly constant (Fig. 5c, *bottom*), causing the mean response to increase with $g$ (Fig. 5e, *bottom*).

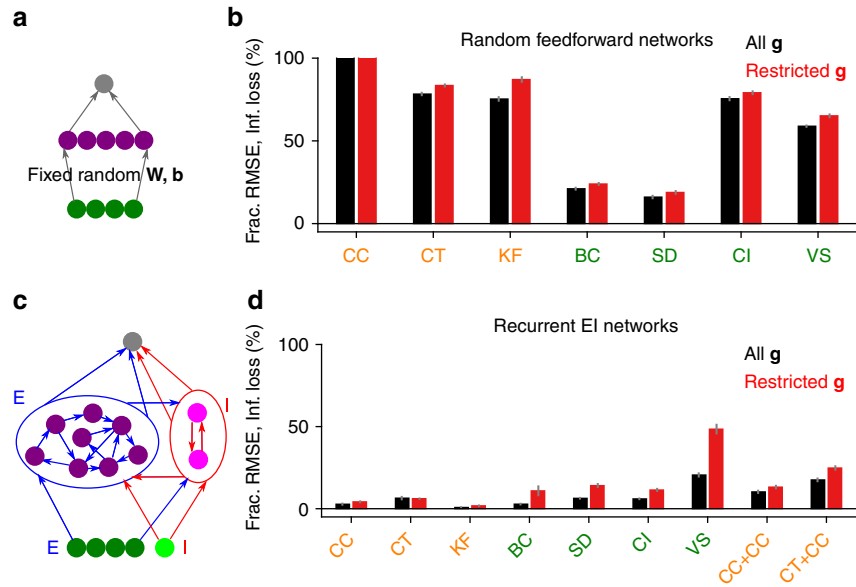

**Fig. 8** Architectural constraints on near-optimal probabilistic inference. **a** Schematic diagram of a random feedforward network, where the input-to-hidden layer weights were set randomly and fixed. **b** Performance of random feedforward networks. The fractional RMSEs in the CC task are over 100%. The *y* axis is cut off at 100%. Random feedforward networks perform substantially worse than their fully trained counterparts. **c** Schematic diagram of a recurrent EI network that obeys Dale's law: inhibitory connections are represented by the *red arrows* and excitatory connections by the *blue arrows*; inhibitory neurons are represented by *lighter colors*, excitatory neurons by *darker colors*. Both input and hidden layers are divided into excitatory-inhibitory subpopulations at a 4-to-1 ratio. Inputs to the network were presented over 10 time steps. The network's estimate was obtained from its output at the final time point. **d** Performance of recurrent EI networks. Introducing more biological realism does not substantially reduce the performance. *Error bars* in **b** and **d** represent standard errors over 10 independent runs of the simulations

On the basis of our simple mean-field model, we therefore predicted that for those tasks where the net input to the output unit is approximately *g*-invariant, there should be a positive correlation between the sparsity of hidden unit responses and the input gain and no (or only a weak) correlation between the mean hidden unit response and the input gain. On the other hand, in tasks such as causal inference, where the net input to the output unit has a strong *g*-dependence, we predicted a positive correlation between the mean hidden unit response and the input gain and no (or only a weak) correlation between the sparsity of hidden unit responses and the input gain. We tested these predictions on our trained networks and confirmed that they were indeed correct (Fig. 6a, b). For causal inference, visual search and stimulus demixing tasks, the correlation between the mean input and the sparsity of hidden layer responses was weak (Fig. 6f), whereas for the remaining tasks, it was strong and positive. The opposite pattern was seen for the correlation between the mean input and the mean hidden layer response (Fig. 6a). In *g*-dependent tasks, such as causal inference, posterior uncertainty is thus represented largely in the mean hidden layer activity; whereas in approximately *g*-invariant tasks, such as coordinate transformation, it is represented largely in the sparsity of hidden layer activity. The sparsity-based representation of posterior uncertainty in *g*-invariant tasks was again driven by large negative mean input-to-hidden layer weights and large positive mean biases (Fig. 6c, d).

The difference between these two types of tasks (*g*-invariant and *g*-dependent) was also reflected in the tuning functions that developed in the hidden layers of the networks. For approximately *g*-invariant tasks, such as coordinate transformation, increasing the input gain *g* sharpens the tuning of the hidden units (Fig. 7a), whereas for *g*-dependent tasks, such as causal inference, input gain acts more like a multiplicative factor scaling the tuning functions without changing their shape (Fig. 7b).

We finally emphasize that these results depend on the linear read-out of hidden layer responses. In continuous tasks, for example, if we use a divisively normalized decoder instead of a linear read-out, posterior uncertainty is no longer encoded in the sparsity of hidden layer responses, but in the mean hidden layer response (Supplementary Fig. 6). Linear read-outs are frequently used in the literature[31–36], hence it is not an unrealistic assumption.

**Random networks**. To investigate the architectural constraints on the networks capable of performing near-optimal probabilistic inference, we considered an alternative architecture, in which the input-to-hidden layer weights and the biases of the hidden units were set randomly and left untrained; only the hidden-to-output layer weights and the biases of the output units were trained (Fig. 8a). Such random networks can be plausible models of some neural systems[37, 38]. Given the same amount of computational resources, these networks performed substantially worse than the fully trained networks (Fig. 8b). A well-known theoretical result can explain the inefficiency of random networks[39]: the approximation error of neural networks with adjustable hidden units scales as $O(1/n)$ with $n$ denoting the number of hidden units, whereas for networks with fixed hidden units, as in our random networks, the scaling is much worse: $O(1/n^{2/d})$, where $d$ is the dimensionality of the problem, suggesting that they need exponentially more neurons than fully trained networks in order to achieve the same level of performance.

**Making the networks biologically more realistic**. So far, we have only considered feedforward networks with undifferentiated neurons. To investigate whether introducing more biological realism would severely constrain the capacity of the networks to perform near-optimal probabilistic inference, following the approach proposed in ref.[40], we trained fully recurrent networks

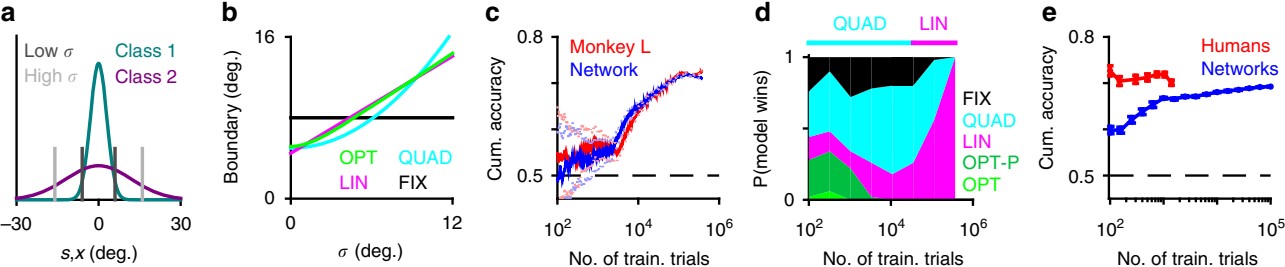

**Fig. 9** Error-based learning in generic neural networks accounts for a monkey subject's performance, but not human subjects' performance in a probabilistic binary categorization task involving arbitrary categories. **a** Structure of the two categories. *Vertical lines* indicate the optimal decision boundaries at low and high noise, σ. **b** Dependence of the decision boundary on sensory noise, σ, for the optimal model (OPT) and three suboptimal heuristic models. **c** Cumulative accuracy of monkey L (*red*) compared with the cumulative accuracy of a neural network trained with the same set of stimuli (*blue*). Cumulative accuracy at trial *t* is defined as the accuracy in all trials up to and including *t*. The network was trained fully on-line. The input noise in the network was matched to the sensory noise estimated for the monkey and the learning rate was optimized to match the monkey's learning curve (see "Methods" section). *Dotted lines* indicate the 95% binomial confidence intervals. **d** The *overbar* shows the winning models for the monkey's behavioral data throughout the course of the experiment according to the AIC metric. The QUAD model initially provides the best account of the data. The LIN model becomes the best model after a certain point during training. The area plot below shows the fractions of winning models, as measured by their AIC scores, over 50 neural network subjects trained with the same set of input noise and learning rate parameters as the one shown in **c**. Similar to the behavioral data, early on in the training, QUAD is the most likely model to win; LIN becomes the most likely model as training progresses. OPT-P is equivalent to an OPT model where the prior probabilities of the categories are allowed to be different from 0.5 (in the experiment, both categories were equally likely). **e** Average performance of six human subjects in the main experiment of ref. [13] (*red*), average performance of 30 neural networks whose input noise was set to the mean sensory noise estimated for the human subjects (*blue*). *Error bars* represent standard errors across subjects. Human and monkey behavioral data in **c** and **e** were obtained from ref. [13]

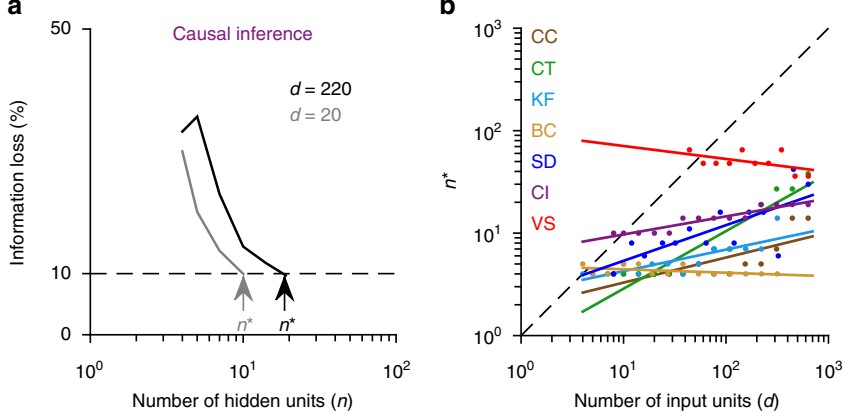

**Fig. 10** Low computational complexity of standard psychophysical tasks and the efficiency of generic networks. **a** For each number of input units, *d*, the minimum number of hidden units required to reach within 10% of optimal performance (15% for visual search), $n^*$, is estimated (shown by the *arrows*). An example is shown here for the causal inference task with *d* = 20 and *d* = 220 input units, respectively. **b** $n^*$ plotted as a function of the total number of input units, *d*, in different tasks. *Solid lines* show linear fits. In these simulations, the number of training trials for each task was set to the maximum number of training trials shown in Supplementary Fig. 1. The growth of $n^*$ with *d* is sublinear in every case

with separate excitatory and inhibitory (EI) populations consisting of rate neurons that respected Dale's law: excitatory neurons projecting only positive weights, inhibitory neurons-only negative weights (see "Methods" section). The input populations were also divided into excitatory and inhibitory subpopulations that obeyed Dale's law (Fig. 8c). The performance of these recurrent EI networks were slightly worse than, but not substantially different from, the corresponding fully trained feedforward networks (Fig. 8d). Moreover, recurrent neurons in these networks recoded sensory reliability in a similar manner to the feedforward networks (Supplementary Fig. 4), suggesting that the main results reported for feedforward networks are robust to the incorporation of more biological realism into our networks.

**Error-based learning accounts for the time course of behavior.** The dependence of the networks' performance on the number of

training trials (Supplementary Fig. 1) suggests a possible explanation for deviations from optimal inference sometimes observed in experimental studies: i.e., insufficient training in the task. Testing this hypothesis rigorously is complicated by possible prior exposure of the subjects to similar stimuli or tasks under natural conditions. Among the tasks considered in this paper, the binary categorization task minimizes such concerns, because it involves classifying stimuli into arbitrary categories. Moreover, in this task, the behavior of both human and monkey observers were best accounted for by heuristic models that were quantitatively suboptimal, but qualitatively consistent with the optimal inference model[13]. Therefore, we sought to test the insufficient-training hypothesis for suboptimal inference in this task.

The stimulus distributions for the two categories and the decision boundaries predicted by the optimal (OPT) and three suboptimal models (FIX, LIN, and QUAD) are shown in

Fig. 9a, b. Different suboptimal models make different assumptions about the dependence of the decision boundary on the sensory noise level, $\sigma$ (Fig. 9b). In particular, the FIX model assumes that the decision boundary is independent of $\sigma$, whereas LIN and QUAD models assume that the decision boundary is a linear or quadratic function of $\sigma$, respectively[13]. The learning curve of a monkey subject who performed a large number of trials in this task (monkey L in ref. [13]) is shown in Fig. 9c together with the performance of a neural network that received the same sequence of trials as the subject. The input noise of the network was matched to the sensory noise estimated for the subject and the learning rate of the network was optimized to fit the learning curve of the subject. The neural network was trained online, updating its parameters after each trial, in an analogous manner to how the monkey subject learned the task.

Besides providing a good fit to the learning curve of the subject (Fig. 9c), the neural networks also correctly predicted the progression of the models that best fit the subject's data, i.e., early on in the training the QUAD model, then the LIN model (Fig. 9d). When we performed the same type of analysis on human subjects' data, human observers consistently outperformed the networks and the networks failed to reproduce the learning curves of the subjects (Fig. 9e). There might be several possible non-exclusive explanations for this finding. First, prior to the experiment, human observers were told about the task, including what examples from each category looked like. This type of knowledge would be difficult to capture with error-based learning alone and might have given human observers a head-start in the task. Second, human observers might have benefited from possible prior familiarity with similar tasks or stimuli. Third, human observers might be endowed with more powerful computational architectures than simple generic neural networks that allow them to learn faster and generalize better[41].

**Efficiency of generic networks**. For each of our tasks, we empirically determined the minimum number of hidden units, $n^*$, required to achieve a given level of performance (15% information loss for visual search, 10% fractional RMSE, or information loss for the other tasks) as a function of the total number of input units, $d$, in our generic networks. An example is shown in Fig. 10a for the causal inference task with $d = 20$ and $d = 220$. The scaling of $n^*$ with $d$ was better than $O(d)$, i.e., sublinear, in all our tasks (Fig. 10b and Supplementary Table 1). Previous theoretical work suggests that this result can be explained by the smoothness properties of the target functions and the efficiency of the generic neural networks with adjustable hidden units. In particular, Barron[39] showed that the optimal number of hidden units in a generic neural network with a single layer of adjustable hidden units scales as $C_f(d)/\sqrt{d}$ with $d$, where $C_f$ is a measure of the smoothness of the target function, with more smooth functions having lower $C_f$ values. As an example, in ref. [39], it was shown that for the $d$-dimensional standard Gaussian function, $C_f$ can be upper-bounded by $\sqrt{d}$, leading to an estimate of $O(1)$ hidden units in terms of $d$. For some of our tasks (for example, binary categorization; Fig. 10b), the scaling of $n^*$ with $d$ was approximately constant over the range of $d$ values tested, suggesting smoothness properties similar to a $d$-dimensional standard Gaussian. For the other tasks, the scaling was slightly worse, but still sublinear in every case: in the worst case of coordinate transformation, linear regression of $\log n^*$ on $\log d$ yields a slope of 0.56 ($R^2 = 0.88$, $P < 10^{-6}$). We can gain some intuition about the relatively benign smoothness properties of our tasks by looking at the analytic expressions for the corresponding target functions (Eqs. (5)–(17)): although the inputs are high dimensional, the solutions can usually be expressed as smooth functions

of a small number of one-dimensional linear projections of the inputs.

The efficiency of our generic networks contrasts sharply with the inefficiency of the manually crafted networks in earlier PPC studies[7, 9–13]: except for the linear cue combination task, these hand-crafted networks used a quadratic expansion, which requires at least $O(d^2)$ hidden units. Moreover, unlike generic neural networks, these networks with hand-crafted hidden units are not guaranteed to work well in the general case, if, for example, the target function is not expressible in terms of a quadratic expansion. Stacking the quadratic expansions hierarchically to make the networks more expressive would make the scaling of the number of hidden units with $d$ even worse (for example, ref. [11]). The fundamental weakness of these hand-crafted networks is the same as that of the random networks reviewed above: they essentially use a fixed basis set theoretically guaranteed to have much worse approximation properties than the adjustable basis of hidden units used in our generic networks[39].

## Discussion

We have shown that small generic neural networks trained with a standard error-based learning rule, but without any explicitly probabilistic feedback or training objective, implement probabilistic inference in simple psychophysical tasks and generalize successfully beyond the conditions they are trained in. Our tasks all assumed psychophysically realistic levels of sensory noise. At these noise levels, simple heuristic non-probabilistic models that do not take trial-to-trial uncertainty into account are unable to mimic the performance of the trained networks and the optimal models.

For tasks with continuous outputs, we trained our networks to minimize the squared error loss function, which is minimized by the posterior mean estimate. Given the universal approximation guarantees for multilayer neural networks with rectified linear hidden units[42], it is not surprising that our networks can approximate the posterior mean given enough hidden units and training data. However, the findings that near-optimal performance can be achieved even in small networks trained with a relatively small number of training examples and that the networks can generalize successfully beyond the training data they receive depend on the particular problems we studied, in particular, on their low-dimensional nature and their smoothness properties, hence are not predicted by the universal approximation results. Moreover, representing the posterior mean is necessary, but not sufficient for general probabilistic computation: it is also necessary to represent uncertainty on a trial-by-trial basis. Using modular tasks, we showed that the networks implicitly represent uncertainty as well, even though the representation of posterior uncertainty is not required for performing the trained task. This finding also holds when the networks are trained with absolute error loss, rather than mean squared error (Supplementary Fig. 5a). In general, we expect higher moments of the posterior to be decodable from non-linear functions of the hidden layer responses, even when the trained task does not require the representation or use of such higher moments. This is likely to be a generic property, since it is known that a small number of random projections of the input have similar information preservation guarantees under general conditions[43, 44].

For tasks with categorical outputs, the output layer of a multilayer neural network is asymptotically guaranteed to converge to the posterior probabilities of the classes under a broad class of loss functions[23], including cross-entropy and mean squared error (Supplementary Fig. 5b). For the particular problems we studied, our results again show empirically that this

convergence can be achieved relatively fast and does not require large networks.

Random networks with only trainable read-out weights performed poorly in our tasks. This can be understood as a consequence of the poor approximation properties of such networks[39]. On the other hand, making our networks biologically more plausible by making them fully recurrent and introducing separate excitatory and inhibitory populations throughout the network that respect Dale's law in their connectivity pattern did not significantly impair the performance of the networks.

Our work is consistent with the PPC framework according to which, by virtue of the variability of their responses, neural populations encode probability distributions rather than single estimates and computations with probability distributions can be carried out by suitable operations on the corresponding neural responses[7, 8]. However, our work disagrees with the existing literature on the implementation of such computations. We show that these computations do not require any special neural operations or network architectures than the very generic ones that researchers have been using for decades in the neural network community[14–16].

The recent literature on PPC respects the principle that Poisson-like neural variability, i.e., exponential-family variability with linear sufficient statistics[7], be preserved between the input and output of a network, because this leads to a fully modular code that can be decoded with the same type of decoder throughout the network[7]. To obtain actual networks, these studies then postulate a literal, one-to-one correspondence between the required neural computations that must be computed at the population level and the computations individual neurons perform. This literal interpretation leads to inefficient neural architectures containing intermediate neurons that are artificially restricted to summing or multiplying the activities of at most two input neurons and that perform substantially different operations in different tasks (for example, linear summation, multiplication, or different forms of divisive normalization)[7, 9–13].

Our generic networks are not necessarily inconsistent with the principle of the preservation of Poisson-like variability between the input and output of a network. Our categorical networks already satisfy this principle, and our continuous networks satisfy it approximately if, instead of a purely linear decoder, we use a linear decoder that is then normalized by the total activity in the hidden layer (Supplementary Fig. 6). However, our results show that it is unnecessary and inefficient to postulate a direct correspondence between population-level and individual-neuron computations: standard neural networks with rectified linear hidden units that perform the same type of operation independent of the task implement population-level computations required for optimal probabilistic inference far more efficiently.

Our results lead to several experimentally testable predictions. First, for gain-invariant tasks, we predict a novel sparsity-based coding of posterior uncertainty in cortical areas close to the behavioral read-out. Stimulus manipulations that increase sensory reliability, such as an increase in contrast of the stimulus, would be expected to increase the sparseness of the population activity in these areas. Another straightforward consequence of this relationship would be a positive correlation between the performance of the animal in the task and the population sparsity of neurons recorded from the same areas. Second, for gain-dependent tasks, such as causal inference, we predict a different coding of posterior uncertainty based on the mean activity in areas close to the read-out. Moreover, based on our mean-field model of the mechanism underlying these two types of codes, we expect a trade-off between them: the stronger the correlation between sparsity and posterior uncertainty, the weaker the relationship between the mean activity and posterior uncertainty

and vice versa. This can be tested with population recordings from multiple areas in multiple tasks. Third, at the level of single cells, we predict tuning curve sharpening with increased input gain in tasks where a sparsity-based coding of reliability is predicted (Fig. 7a). Such tuning curve sharpening has been observed in cortical areas MT[45], MST[46], and MSTd[29]. On the other hand, we expect the input gain to act more like a multiplicative factor in tasks where a mean activity-based coding of reliability is predicted (Fig. 7b).

Sparse and reliable neural responses have been observed under natural stimulation conditions[47–49]. Inhibitory currents have been shown to be crucial in generating such sparse and reliable responses[47, 50], reminiscent of the importance of negative mean input in our mean-field model of the sparsity-based coding of posterior uncertainty (Fig. 5b).

Our networks are highly idealized models of real neural circuits. Although we validated our basic results using biologically more realistic recurrent excitatory-inhibitory networks (Fig. 8c, d), even these networks are simplistic models that ignore much of the complexity of real neural circuits. For example, real neural circuits involve several morphologically and physiologically distinct cell types with different connectivity patterns and with potentially distinct functions[51]. Real neurons also implement a diverse set of complicated non-linearities, unlike the simple rectification nonlinearity we assumed in our networks. It remains to be determined what possible functional roles this diversity plays in neural circuits.

However, even seemingly drastic simplifications can, in some cases, yield insights about the brain. For example, cortical networks are usually highly recurrent, thus modeling them as feedforward networks might seem like an over-simplification. However, networks with feedback connections can sometimes behave effectively like a feedforward network[52, 53]. As another example, feedforward networks also currently provide the best characterization of the neural responses in higher visual cortical areas[17, 20], even though these areas are known to involve abundant feedback connections both within the same area and between different areas. Therefore, insights gained from understanding simplified models can still be relevant for understanding real cortical circuits.

Second, our networks were trained with the backpropagation algorithm, which is usually considered to be biologically unrealistic due to its non-locality. Although the backpropagation algorithm in its standard form, we have implemented is indeed biologically unrealistic, biologically plausible approximations, or alternatives, to backpropagation have been put forward recently[54, 55]. Therefore, it is quite likely that one need not compromise the power of backpropagation in order to attain biological plausibility.

Third, the stimuli that we used were far from naturalistic. However, the computations required in our tasks capture the essential aspects of the computations that would be required in similar tasks with natural stimuli. Using simple stimuli allows for the parametric characterization of behavior and makes the derivation of the optimal solution more tractable. We have shown here that new insights can be obtained by combining analytically derived optimal solutions with neural networks. For example, understanding the novel sparsity-based representation of posterior uncertainty in the hidden layers of the networks in some tasks but not in others relied on the analysis of the optimal solutions in different tasks.

Finally, as exemplified by the inability of error-based learning to account for the performance of human observers in the binary categorization task, we do not expect error-based learning in generic neural networks to fully account for all aspects of the performance of human observers, and possibly non-human

observers as well, even in simple tasks. Relatively mundane manipulations, such as changing the target or the distractors, or the number of distractors in a visual search task, changing the duration of a delay interval in a short-term memory task require wholesale retraining of generic neural networks, which seems to be inconsistent with the way human observers, and possibly non-human observers, can effortlessly generalize over such variables. More powerful architectures that combine a neural network controller with an external memory can both learn faster and generalize better[41], offering a promising direction for modeling the generalization patterns of observers in simple psychophysical tasks.

## Methods

**Neural networks.** In all networks, the input units were independent Poisson neurons: $\mathbf{r}_{in} \sim \text{Poisson}(\mathbf{f}(s, c))$, where $\mathbf{f}$ is the vector of mean responses (tuning functions), $s$ is the stimulus, and $c$ is a stimulus contrast or coherence variable that controls the quality of sensory information. For the main results presented in the paper, we assume that the effect of $c$ can be described as a multiplicative gain scaling: $\mathbf{f}(s, c) = g(c)\mathbf{f}(s)$, where the individual tuning functions comprising $\mathbf{f}(s)$ were either linear (stimulus demixing), von Mises (visual search), or Gaussian (all other tasks).

To demonstrate the generality of our approach, we also considered two alternative ways in which stimulus contrast or coherence can affect the responses of the input population. In particular, for the cue combination task, we considered tuning functions, where $c$ did not have a purely multiplicative effect, but affected the baseline responses as well[28]: $f(s, c) = cf(s) + (1 - c)\beta$ with $0 \le c \le 1$, where $\beta$ was chosen such that the mean response of the input population was independent of $c$. Second, again for the cue combination task with two cues, we considered tuning functions where stimulus contrast $c$ affected both the peak response and the preferred stimuli of input neurons, as reported in ref. [30] for speed tuning in area MT: $f(s, c) = r_0 + Ag(c)\exp\left(-\frac{1}{2\sigma^2}\left(\log\frac{s+s_0}{Bg(c)\phi+s_0}\right)\right)$ with the following parameters: $r_0 = 0.5$, $A = 5$, $s_0 = 1$, $\sigma = 1$, $B = 10$, and $g(c) = 1/((\alpha c)^{-\beta} + \gamma)$ with $\alpha = 10$, $\beta = 2$, $\gamma = 3$. The results for these two cases are shown in Fig. 4c.

The hidden units in both feedforward and recurrent networks were ReLUs. In feedforward networks, the hidden unit responses are described by the equation: $\mathbf{r}_{hid} = [\mathbf{W}_{in}\mathbf{r}_{in} + \mathbf{b}]_+$ and in recurrent networks by the equation: $\mathbf{r}_{hid,t+1} = [\mathbf{W}_{in}\mathbf{r}_{in,t+1} + \mathbf{W}_{rec}\mathbf{r}_{hid,t} + \mathbf{b}]_+$, where $\mathbf{W}_{in}$ and $\mathbf{W}_{rec}$ are the input and recurrent weights, respectively, and $[\cdot]_+$ denotes elementwise rectification. For tasks with continuous output variables, the network output corresponds to a linear combination of the hidden unit responses: $y = \mathbf{w}^T\mathbf{r}_{hid} + b$, and in tasks with categorical variables, the network output was given by a linear combination of the hidden unit responses passed through a sigmoid nonlinearity: $y = \sigma(\mathbf{w}^T\mathbf{r}_{hid} + b)$.

In the recurrent EI networks, all connections were constrained to satisfy Dale's law as in ref. [40]. The recurrent units are all rate-based neurons. The ratio of excitatory to inhibitory neurons in both input and recurrent populations was 4:1. Inputs were presented over 10 time steps, but the total input information was equated to the total input information in the feedforward networks. The network's estimate was obtained from its output at the final, i.e., 10th time step. Other details are the same as in the corresponding simulations in the feedforward case.

In random networks, input-to-hidden layer weights were sampled from a normal distribution with zero mean and standard deviation $\sqrt{2/(n + d)}$, where $d$ is the number of input neurons and $n$ is the number of hidden neurons[56]; biases of the hidden units were sampled from a normal distribution with zero mean and standard deviation of 0.1.

In the main simulations, the networks had 200 hidden units. In cue combination, modular cue combination, coordinate transformation, Kalman filtering, binary categorization, and causal inference tasks, there were 50 input neurons per input population. To make our results comparable to earlier results, we used 20 input neurons per input population in the visual search task and 10 input neurons per input population in the stimulus demixing task.

**Non-probabilistic models.** For the Kalman filtering and all cue combination tasks, we used heuristic, non-probabilistic reference models that estimated the individual cues (and the past state in Kalman filtering) optimally, but combined them suboptimally by weighting them equally regardless of their reliability. As indicated before, these models still performed a non-trivial probabilistic computation, namely marginalizing out a nuisance variable, i.e., the input gain, to come up with the optimal estimate of the individual cues. We also note that for the coordinate transformation task, unlike in cue combination or Kalman filtering, the optimal combination rule does not depend on the reliabilities of the individual inputs. For the categorical tasks, we also used non-probabilistic reference models that assumed equal reliabilities for individual inputs. To give the reference models the best chance to perform well, we chose the assumed common reliability of the inputs that minimized the information loss.

**Training procedure.** The feedforward networks were trained with the standard backpropagation algorithm[14]. The recurrent networks were trained with backpropagation through time[15]. We used the Adam stochastic gradient descent algorithm[57] to implement backpropagation. The batch sizes for the updates were 10 for binary categorization, 500 for visual search, and 100 for the other tasks. In Fig. 9, we used an online vanilla stochastic gradient descent algorithm with learning rate decrease over trials described by $\eta_0/(1 + \gamma t)$, where $t$ is the trial number. The parameters $\eta_0$ and $\gamma$ were fit to the monkey's learning curve.

**Training conditions.** The "all $\mathbf{g}$" conditions in different tasks were as follows. In cue combination and coordinate transformation tasks, all 25 pairs of the form $(g_1, g_2)$ with $g_1, g_2 \in \{0.25, 0.5, 0.75, 1, 1.25\}$ were presented an equal number of times. In Kalman filtering, $g$ was uniformly drawn between 0.3 and 3 at each time step. In binary categorization, the six gain values, $g \in \{0.148, 0.36, 0.724, 1.128, 1.428, 1.6\}$, were presented an equal number of times. These gain values were calculated from the mean noise parameter values reported for the human subjects in ref. [13]. In causal inference, all 25 pairs of the form $(g_1, g_2)$ with $g_1, g_2 \in \{0.5, 1, 1.5, 2, 2.5\}$ were presented an equal number of times. In stimulus demixing, following[10], $c$ was uniformly and independently drawn between 2 and 9 for each source. In visual search, $g$ was randomly and independently set to either 0.5 or to 3 for each stimulus.

The "restricted $\mathbf{g}$" conditions in different tasks were as follows. In cue combination and coordinate transformation tasks, the two pairs $(g_1, g_2) \in \{(0.25, 0.25), (1.25, 1.25)\}$ were presented an equal number of times. In Kalman filtering, $g$ was randomly and independently set to either 0.3 or to 3 at each time step. In binary categorization, $g$ was always 1.68. This gain value corresponds to 100% contrast as calculated from the mean noise parameter values for the human subjects reported in ref. [13]. In causal inference, pairs of the form $(g_1, g_2) \in \{(0.5, 0.5), (2.5, 2.5)\}$ were presented an equal number of times. In stimulus demixing, $c$ was either set to 2 for all sources or else set to 9 for all sources. Similarly, in visual search, $g$ was either set to 0.5 for all stimuli or else set to 3 for all stimuli.

**Mean-field model of hidden unit responses.** For a given input activity $\mathbf{r}$, we consider the responses of the hidden units as realizations of a random variable $r_{hid}$. The output weights are also assumed to be realizations of a random variable $w$. We further assume that $w$ and $r_{hid}$ are independent. The network's output is then proportional to $\langle w \rangle \langle r_{hid} \rangle$. We want to make this expression invariant to the input gain $g$. We first introduce a measure of the total sensitivity of this expression to variations in $g$. We will do this by computing the magnitude of the derivative of $\langle w \rangle \langle r_{hid} \rangle$ with respect to $g$ and integrating over a range of $g$ values, but we first note that the output weights are already gain invariant, hence we can just consider $\langle r_{hid} \rangle$. We now have to find an expression for $\langle r_{hid} \rangle$. The net input to a typical hidden unit is given by:

$$g\mathbf{w}_{in}^T\mathbf{r} + b \sim \mathcal{N}\left(\mu_* \equiv g\mu + \mu_b, \sigma_*^2 \equiv g^2\sigma^2 + \sigma_b^2\right) \quad (1)$$

where $\mathbf{w}_{in}$ are the input weights to a typical hidden unit. Then:

$$\bar{\mu} \equiv \langle r_{hid} \rangle = \left\langle \left[g\mathbf{w}_{in}^T\mathbf{r} + b\right]_+ \right\rangle = \left[1 - \Phi\left(\frac{-\mu_*}{\sigma_*}\right)\right]\mu_* + \phi\left(\frac{-\mu_*}{\sigma_*}\right)\sigma_* \quad (2)$$

where $\Phi(\cdot)$ and $\phi(\cdot)$ are the cdf and the pdf of the standard Gaussian distribution. As mentioned above, we then introduce the following measure of the total sensitivity of $\bar{\mu}$ to variations in $g$:

$$T_{var} = \int_{g_{min}}^{g_{max}} |\bar{\mu}'(g)| \, dg \quad (3)$$

where the prime represents the derivative with respect to $g$. Because $g$ always appears as $g\mu$ or $g\sigma$ in $\bar{\mu}$ (Eq. 2), the parametrization in terms of $g$, $\mu$, and $\sigma$ is redundant. We therefore set $\sigma = 1$, and hence expressed everything in terms of the scale of $\sigma$. We then minimized $T_{var}$ numerically with respect to $\mu$, $\mu_b$, and $\sigma_b$ subject to the constraint that the mean response across different gains be equal to some positive constant $K$:

$$\frac{1}{g_{max} - g_{min}} \int_{g_{min}}^{g_{max}} \bar{\mu}(g) \, dg = K > 0 \quad (4)$$

This ensures that the degenerate solution where the hidden layer is completely silent is avoided.

**Task details.** In the linear cue combination task, the objective is to combine two cues, $\mathbf{r}_1$ and $\mathbf{r}_2$, encoding information about the same variable, $s$, in a statistically optimal way. Assuming a squared error loss function, this can be achieved by computing the mean of the posterior $p(s|\mathbf{r}_1, \mathbf{r}_2)$. For a uniform prior distribution, the posterior mean is given by an expression of the form[7]:

$$\hat{s}_{opt} = \frac{\phi^T(\mathbf{r}_1 + \mathbf{r}_2)}{\mathbf{1}^T(\mathbf{r}_1 + \mathbf{r}_2)}, \quad (5)$$

where $\phi$ is the vector of preferred stimuli of input neurons and $\mathbf{1}$ is a vector of ones.

This expression is approximate when the prior is not uniform over the entire real line and the quality of the approximation can become particularly bad in the high-noise regime considered in this paper. Thus, in practice, we computed posterior means numerically, rather than using the above equation. The equation is still useful, however, in helping us understand the type of computation the network needs to perform to approximate optimal probabilistic inference. During training, the two cues received by the input populations were always non-conflicting: $s_1 = s_2 = s$ and the gains of the input populations varied from trial to trial. The network was trained to minimize the mean squared error between its output and the common $s$ indicated by the two cues.

In the coordinate transformation task, the eye-centered location of an object in 1-d, $s_1$, is encoded in a population of Poisson neurons with responses $\mathbf{r}_1$ and the current eye position, $s_2$, is similarly encoded in a population of Poisson neurons with responses $\mathbf{r}_2$. The goal is to compute the head-centered location of the object, which is given by $s = s_1 + s_2$. Assuming uniform priors, the optimal estimate of $s$ can be expressed as[10]:

$$\hat{s}_{\text{opt}} = \frac{\mathbf{r}_1^{\mathsf{T}} \mathbf{B} \mathbf{r}_2}{\mathbf{r}_1^{\mathsf{T}} \mathbf{A} \mathbf{r}_2} \tag{6}$$

for suitable matrices $\mathbf{B}$ and $\mathbf{A}$ (see ref. [10] for a full derivation). Again, when the priors are not uniform over the real line, this expression becomes only approximate and posterior means are computed numerically.

In the Kalman filtering task, we considered a one-dimensional time-varying signal evolving according to: $s_t = (1 - \gamma)s_{t-1} + \eta_t$, where $\eta_t \sim \mathcal{N}\left(0, \sigma_\eta^2\right)$ with $\gamma = 0.1$ and $\sigma_\eta^2 = 1$. At each time $t$, the stimulus was represented by the noisy responses, $\mathbf{r}_{\text{in},t}$, of a population of input neurons with Poisson variability. The input population projected to a recurrent pool of neurons that have to integrate the momentary sensory information coming from the input population with an estimate of the signal at the previous time step (as well as the uncertainty associated with that estimate) to perform optimal estimation of the signal at the current time step. We decoded the estimate of the signal at each time step by a linear read-out of the recurrent pool: $\hat{s}_t = \mathbf{w}^{\mathsf{T}} \mathbf{r}_{\text{rec},t} + b$. The network was trained with sequences of length 25 using a squared error loss function. The posterior $p(s_t|\mathbf{r}_{\text{in, 1:}t})$ is Gaussian with natural parameters given recursively by[10]:

$$\frac{\mu_t}{\sigma_t^2} = \frac{\mu_{\text{in},t}}{\sigma_{\text{in},t}^2} + \frac{(1 - \gamma)\mu_{t-1}}{(1 - \gamma)^2 \sigma_{t-1}^2 + \sigma_\eta^2} \tag{7}$$

$$\frac{1}{\sigma_t^2} = \frac{1}{\sigma_{\text{in},t}^2} + \frac{1}{(1 - \gamma)^2 \sigma_{t-1}^2 + \sigma_\eta^2} \tag{8}$$

where $\mu_{\text{in},t}$ and $\sigma_{\text{in},t}^2$ are the mean and variance of $p(s_t|\mathbf{r}_{\text{in},t})$, which represents the momentary sensory evidence encoded in the input population. These are, in turn, given by $\mu_{\text{in},t} = \phi^{\mathsf{T}} \mathbf{r}_{\text{in},t} / \mathbf{1}^{\mathsf{T}} \mathbf{r}_{\text{in},t}$ and $\sigma_{\text{in},t}^2 = \sigma_f^2 / \mathbf{1}^{\mathsf{T}} \mathbf{r}_{\text{in},t}$.

In the binary categorization task, the goal is to classify a noisy orientation measurement into one of two overlapping classes that have the same mean but different variances. Given a noisy activity pattern $\mathbf{r}$ over the input population representing the observed orientation, the posterior probabilities of the two classes can be calculated analytically. The log-likelihood ratio of the two categories is given by[13]:

$$d \equiv \log \frac{p(\mathbf{r}|C=1)}{p(\mathbf{r}|C=2)} = \frac{1}{2} \left( \log \frac{1 + \sigma_2^2 \mathbf{a}^{\mathsf{T}} \mathbf{r}}{1 + \sigma_1^2 \mathbf{a}^{\mathsf{T}} \mathbf{r}} - \frac{(\sigma_2^2 - \sigma_1^2)(\mathbf{e}^{\mathsf{T}} \mathbf{r})^2}{(1 + \sigma_1^2 \mathbf{a}^{\mathsf{T}} \mathbf{r})(1 + \sigma_2^2 \mathbf{a}^{\mathsf{T}} \mathbf{r})} \right) \tag{9}$$

where $\mathbf{e} = \phi / \sigma_f^2$ and $\mathbf{a} = \mathbf{1} / \sigma_f^2$. The posterior probability of the first class is then given by a sigmoidal function of $d$: $p(C = 1|\mathbf{r}) = 1/(1 + \exp(-d))$.

In the causal inference task, the goal is to infer whether two sensory measurements are caused by a common source or by two separate sources. The log-likelihood ratio of these two hypotheses is given by[12]:

$$d = \frac{z_{11} z_{21}}{z_{12} + z_{22} + J_s} - \frac{1}{2} \left[ \frac{z_{22} z_{11}^2}{(z_{12} + J_s)(z_{12} + z_{22} + J_s)} \right. \\ \left. + \frac{z_{12} z_{21}^2}{(z_{22} + J_s)(z_{12} + z_{22} + J_s)} - \log\left(1 + \frac{z_{12} z_{22}}{J_s(z_{12} + z_{22} + J_s)}\right) \right] \tag{10}$$

where $J_s$ is the precision of the Gaussian stimulus distribution and:

$$\sigma_f^2 z_{11} = \phi_1^{\mathsf{T}} \mathbf{r}_1 \tag{11}$$

$$\sigma_f^2 z_{12} = \mathbf{1}^{\mathsf{T}} \mathbf{r}_1 \tag{12}$$

$$\sigma_f^2 z_{21} = \phi_2^{\mathsf{T}} \mathbf{r}_2 \tag{13}$$

$$\sigma_f^2 z_{22} = \mathbf{1}^{\mathsf{T}} \mathbf{r}_2, \tag{14}$$

where $\sigma_f^2$ is the common variance of the Gaussian tuning functions of the individual input neurons. $\phi_1$ and $\phi_2$ are the preferred stimuli of the neurons in the first and second populations, respectively. For convenience, we assumed $\phi_1 = \phi_2$. The optimal probability of reporting "same cause" is then simply given by $p(C = 1|\mathbf{r}_1, \mathbf{r}_2) = 1/(1 + \exp(-d))$.

In the stimulus demixing task, the goal is to infer the presence or absence of different signal sources in a mixture of signals with unknown concentrations. As a concrete example, the signals can be thought of as different odors, and the task would then be to infer the presence or absence of different odors in an odor mixture with unknown concentrations[10]. Following[10], we assumed a linear mixing model:

$$o_i = \sum_k w_{ik} c_k s_k \tag{15}$$

where $s_k$ denotes the presence or absence of the $k$-th odor source, $c_k$ denotes its concentration, $o_i$ is the concentration of the $i$-th odorant, and $w_{ik}$ is the weight of the $k$-th odor source in the $i$-th odorant. The task can then be formalized as the computation of the posterior probability of the presence or absence of each odor source, given noisy responses $\mathbf{r} = \{\mathbf{r}_i\}_{i=1}^{n_o}$ of populations of Poisson neurons encoding the odorants: i.e., $p(s_k = 1|\mathbf{r})$. The input populations were assumed to have linear tuning for the odorants: $\mathbf{r}_i \sim \text{Poisson}(o_i \mathbf{f}_i + \mathbf{b}_i)$, where $\mathbf{f}_i$ and $\mathbf{b}_i$ were random vectors with positive entries[10]. As in ref. [10], we assumed four sources and four odorants. The networks were trained to minimize the cross-entropy between the network's outputs and the correct source present/absent labels, $s_k$.

In the visual search task, the goal is to infer the presence or absence of a target stimulus $s_T$ among a set of heterogeneous distractors. The log-likelihood ratio of the target presence is given by[11]:

$$d = \log \frac{1}{N} \sum_{i=1}^{N} \exp(d_i) \tag{16}$$

where $N$ is the number of stimuli on the display (we assumed $N = 4$) and the local target presence log-likelihoods $d_i$ are given by:

$$d_i = \mathbf{h}_i(s_T)^{\mathsf{T}} \mathbf{r}_i - \log\left(\frac{1}{\pi} \int_0^\pi \exp\left(\mathbf{h}_i(s_i)^{\mathsf{T}} \mathbf{r}_i\right) ds_i\right) \tag{17}$$

For independent Poisson neurons, the stimulus kernel $\mathbf{h}(s)$ is given by $\mathbf{h}(s) = \log \mathbf{f}(s)$, where we assumed von Mises tuning functions for individual input neurons. The integral in the second term on the right hand side was calculated numerically.

**Behavioral data**. Human and monkey behavioral data used in Fig. 9 were obtained from a previously published study[13]. Human behavioral data reported in Fig. 9e are from six human subjects who completed the main experiment in ref. [13]. The behavioral data reported in Fig. 9c are from monkey L. Only incomplete behavioral data from another monkey that completed the experiment (monkey A) were available. Because data from all trials are needed to obtain a reliable estimate of the subject's learning curve, data from this monkey were not used in the current paper. For further details about the experimental settings, subjects, and model fitting, see ref. [13].

**Code availability**. The code to reproduce the results reported in this paper is available at the following public repository: https://github.com/eminorhan/inevitable-probability.

**Data availability**. The behavioral data reported in Fig. 9 are available at the following public repository: https://github.com/eminorhan/inevitable-probability.

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

## Acknowledgements

This work was supported by Grant R01EY020958 from the National Eye Institute. We thank Edgar Walker for providing us with the monkey behavioral data analyzed in this paper.

## Author contributions

A.E.O. and W.J.M. designed the research; A.E.O. implemented the simulations; A.E.O. and W.J.M. analyzed the results; A.E.O. and W.J.M. wrote the paper.

## Additional information

**Competing interests:** The authors declare no competing financial interests.

