## [Peer Review File · Nature Communications]

PEER REVIEW FILE

Reviewers' Comments:

Reviewer #1 (Remarks to the Author):

This revision addresses my main concern regarding the validity of the results by making it explicit how the hierarchical cue-integration task tests for the implicit representation of the posterior. I think this is a cool idea with potentially many applications in related future work. Together with the innovative and, I think important, approach of exploring probabilistic inference using classically trained feedforward deterministic networks, I think the manuscript presents important and timely contributions to the field.

Unfortunately, the revision does not address my main concern regarding presentation. It is still written as a survey of many tasks for whom the shown results are claimed to support the author's main conclusion without explaining any one task in detail. Even for the very first task, (cue integration), and the very first Results panel (Fig. 2a left), the reader is not completely clear on what is shown. What I would expect is a comparison of the match between network estimate and optimal output for several gain combinations, demonstrating that the network takes relative reliability correctly into account. Importantly, it should include a $g_1=0$, and a $g_1=g_2$ condition. Instead, we are shown an overall comparison where the difference between g_1 and g_2 (is there even one?) is not separated out, and where it's not clear what the errorbars are taken over. Currently, we simply don't have the necessary information to interpret the Figure and decide whether it supports the claim that probabilistic inference is performed or not! Furthermore, it's irritating to be confronted with arbitrary-seeming $-20...20$ x-axis labels, without units, that differ from panel to panel.

As I wrote in my last comments: "On a revision, I think it would also be beneficial to reorganize the manuscript to make it clearer and more easily accessible. I'd suggest starting with 1 representative task, explaining it, the modeling, and the results, in detail. Based on that, move on to the other tasks, only explaining how they differ in the important insights (if at all). I found the current presentation of everything mixed/at the same time confusing."

-> Author Response: This is a very helpful didactic suggestion. The results section has been completely rewritten in the revision to make it more easily accessible.

However, the revision has ignored this suggestion.

I further note that the current presentation relies on the use of something equivalent to "see below" three(!) times over the first few pages, without specifying where exactly.

This manuscript contains results that are worthy of two separate papers; one paper that really digs into 1 or 2 specific tasks, e.g. cue integration and a categorical task, and presents in detail what the network has learnt, and how the posterior is implicitly represented/how it can be read-out. The 2nd paper could then survey a range of other tasks to establish that these findings are general, and not specific to the chosen tasks. Even if all these results are presented in a single paper, I think it's essential to do the first part. At the moment, we have to believe the authors rather than being able to convince ourselves of their main claims based on the figures and analyses shown.

More specific points below:

- why is gain an appropriate reliability dial? makes sense in context of PPC, but is this an important limitation here?

-> Would be good to start by saying that you first consider this case, and later discuss other cases.

- You end up concluding that your network works regardless of how reliability is encoded but that appears too strong since you haven't considered qualitatively different schemes, like different neurons representing parameters like stimulus mean and variance, or neural sampling based codes.

- Also, how does this generalize to multidimensional inputs?

- non-prob reference model not explained at all in Results

- results in Fig 2 shown without explaining details of the task

- Fig 2a: units? why different ranges on x axis?

- assumes these tasks are totally generic

- gain ranges explained too late, needed here!

- what g-ratio used for 2a results? ratio=0, and =1 of particular interest

- too much explanation of what is being done, rather than why

Ex. "For the linear cue combination and coordinate transformation tasks, to test for the representation of posterior uncertainty in the hidden layer, we plugged the trained hidden layers into a network incorporating an additional input population and fixed their parameters"

- Fig 4a and main text: "Note that we use relatively high gains in this particular example to make the optimal combination rule approximately linear. In the low gain conditions used in the main

simulations, the optimal combination rule is no longer linear." This seems to be an important point that I don't understand. Please unpack.

- "We first note that in tasks with continuous output variables, the optimal solution is invariant to a multiplicative scaling of the input responses (Methods, Equations 5-8)."

-> You state it as if it were a law of nature, whereas it's simply how you have set up your input representations.

"We finally note that these results are decoder dependent. In the continuous tasks, for example, if we use a divisively normalized decoder instead of a linear read-out, posterior uncertainty is encoded in the mean hidden layer response, rather than in the sparsity of hidden layer responses (supplementary Figure S4)."

-> Doesn't this imply that little meaning can be attached to this analysis since we simply don't know what the appropriate decoder is?

- Section on recurrent EI-networks potentially very interesting but totally insufficiently explained. Neither in the Results, nor in the Methods does it even say whether these are spiking neurons (LIF?), or binary units. Raises many more questions than it answers and probably best left for another paper.

- Discussion claims: "Our categorical networks already satisfy this principle, and our continuous networks satisfy it if, instead of a purely linear decoder, we use a linear decoder that is then normalized by the total activity in the hidden layer (supplementary Figure S4)"

-> Where is it shown that intermediate and output layers follow Poisson statistics?

- 3rd empirical prediction: how can the dependence depend on the task? Let's say the animal has learnt both tasks and either switches between them or performs them concurrently. How could the responses be task-dependent?

Reviewer #2 (Remarks to the Author):

I think the authors did a great job addressing my concerns, and I'm excited to see how the paper is received by a broader audience.

Reviewer #3 (Remarks to the Author):

I appreciate the effort that the authors have put into revising their paper, and the revised version is an improvement over the previous version of the manuscript. The more detailed elucidation of how these results go beyond previous work, the addition of simulations using other loss

functions in the continuous case, and the inclusion of more biologically realistic neural network models are all things that strengthen the paper. However, for me the two issues that motivated my original review remain: this paper could either be a theoretical paper showing the properties of abstract neural network models, or a neuroscience paper connecting ideas about these models to how the brain works. I still feel like the contributions fall short of the mark under either of these construals.

As noted in the paper, the key theoretical contribution is the demonstration that the correspondence with posteriors is robust across training methods, generalization, and different paradigms. However, I still feel that this is a relatively incremental contribution that wouldn't come as a surprise to most researchers working with neural networks. As the authors note, similar robustness results were already known to hold for the discrete case.

This leaves things to turn on the contributions to neuroscience, and despite the addition of the new simulations these connections are speculative at best. I continue to feel that there's a nice paper here, but one that turns on deepening the connections to neuroscience rather than emphasizing the theoretical results.

Reviewers' comments are highlighted in blue, responses in black.

Reviewer 1

This revision addresses my main concern regarding the validity of the results by making it explicit how the hierarchical cue-integration task tests for the implicit representation of the posterior. I think this is a cool idea with potentially many applications in related future work. Together with the innovative and, I think important, approach of exploring probabilistic inference using classically trained feedforward deterministic networks, I think the manuscript presents important and timely contributions to the field.

We thank the reviewer for their encouraging words.

Unfortunately, the revision does not address my main concern regarding presentation. It is still written as a survey of many tasks for whom the shown results are claimed to support the author's main conclusion without explaining any one task in detail.

[...] *(jumping ahead because the following comment is best discussed along with the previous)*

As I wrote in my last comments: "On a revision, I think it would also be beneficial to reorganize the manuscript to make it clearer and more easily accessible. I'd suggest starting with 1 representative task, explaining it, the modeling, and the results, in detail. Based on that, move on to the other tasks, only explaining how they differ in the important insights (if at all). I found the current presentation of everything mixed/at the same time confusing."

We thank the reviewer for these suggestions regarding presentation and we are sorry for not fully addressing them in our first revision. In the old version, we didn't have any figure explaining the task structure, training and testing conditions, and the tuning curves of the input populations. In the revision, we added new panels in Figure 1 (Figure 1a-b; reproduced below) describing these details and explained them in more detail in the main text.

We hope that this figure will go a long way toward addressing the reviewer's concerns regarding presentation.

We are reluctant to devote a separate figure in the main text to a single task, because (i) this could create the mistaken impression that there is something special about that one task and (ii) we think that the results for individual tasks are quite straightforward and don't deserve to be dwelt upon at length. Therefore, we have decided to keep the network diagrams and results for all tasks in single figures. This presentation also helps to convey one of our key points: previous studies have used hand-crafted, task-specific networks, whereas we use essentially the same network across all tasks.

Even for the very first task, (cue integration), and the very first Results panel (Fig. 2a left), the reader is not completely clear on what is shown. What I would expect is a comparison of the match between network estimate and optimal output for several gain combinations, demonstrating that the network takes relative reliability correctly into account. Importantly, it should include a $g_1=0$, and a $g_1=g_2$ condition.

We are in fact doing what the reviewer expects, although we agree that our presentation could have been improved. On page 3 of the old version, we already wrote (relevant part boldfaced):

We manipulated sensory reliability trial-by-trial via gain variables g multiplying the responses of the input populations, with higher gains corresponding to more reliable sensory information. In each task, networks were tested **with a wide range of gains or gain combinations** (in tasks with more than a single stimulus). To test the generalization capacity of the networks, we trained them with a limited range of gains or gain combinations, as well as with the full range of test gains or gain combinations. The latter unrestricted training regime is called the “All g ” condition, whereas the former limited training regime is called the “Restricted g ” condition in what follows. **The specific gain ranges and gain combinations used in each task are indicated below (Methods).**

In the revision, we added a panel in Figure 1 (Figure 1b) illustrating these training and testing conditions pictorially. The $g_1=0$ condition is not a cue combination task so we did not include it. The $g_1=g_2$ conditions are included, but there is nothing special about these particular combinations. As indicated in the caption (“Optimal estimates vs. the network outputs in “all g ” conditions of the continuous tasks”), Figure 2a-b contains all gain combinations (with up to 5-fold gain difference between g_1 and g_2 in both directions: $g_1>g_2$ and $g_2>g_1$). Color coding the gain combinations did not seem particularly informative to us, since there are a total of 25 gain combinations. Moreover, the small error bars and the very low information loss (both computed across all gain combinations) indicate that near-optimality is reached across gain combinations. To address the reviewer’s concerns and to confirm these observations more directly, we added two supplementary figures to the revised manuscript (supplementary Figure S2-S3; reproduced below) plotting the estimates of the trained networks against optimal estimates in each of the 25 gain combinations in the cue combination and coordinate transformation tasks, respectively, and demonstrating that the networks perform consistently well across all gain combinations.

Instead, we are shown an overall comparison where the difference between g_1 and g_2 (is there even one?) is not separated out, and where it's not clear what the errorbars are taken over.

Error bars are standard deviations over trials. We added this information to the caption of Figure 2.

Currently, we simply don't have the necessary information to interpret the Figure and decide whether it supports the claim that probabilistic inference is performed or not! Furthermore, it's irritating to be confronted with arbitrary-seeming $-20 \dots 20$ x-axis labels, without units, that differ from panel to panel.

We made the stimulus range consistent across figures by normalizing to $[-1, 1]$. However, the units are arbitrary – they represent arbitrary linear ranges. We indicated this by adding “a.u.” in the revised figures.

I further note that the current presentation relies on the use of something equivalent to “see below” three (!) times over the first few pages, without specifying where exactly.

In the instances that we could find, we now specify where exactly; some of these references are still “see Methods” but we believe this is an acceptable and widely used practice.

This manuscript contains results that are worthy of two separate papers; one paper that really digs into 1 or 2 specific tasks, e.g. cue integration and a categorical task, and presents in detail what the network has learnt, and how the posterior is implicitly represented/how it can be read-out. The 2nd paper could then survey a range of other tasks to establish that these findings are general, and not specific to the chosen tasks. Even if all these results are presented in a single paper, I think it's essential to do the first part. At the moment, we have to believe the authors rather than being able to convince ourselves of their main claims based on the figures and analyses shown.

We are reluctant to divide our paper into two separate papers. As mentioned before, we think that the results from individual tasks are quite straightforward and hence would not justify a separate paper in our opinion. Our results hang together quite naturally, and most of our key results (specifically the sparsity-based representation of posterior uncertainty in different tasks, the efficiency of the networks in different tasks) rely on being able to analyze multiple tasks.

- why is gain an appropriate reliability dial? makes sense in context of PPC, but is this an important limitation here?

-> Would be good to start by saying that you first consider this case, and later discuss other cases.

Good suggestion. We indeed considered alternative “reliability dials” other than purely multiplicative gain as discussed on page 6 and Figure 4b-c. We rewrote the discussion around page 3 to say that these alternative schemes are considered later in the paper: “For the main

experiments, we manipulated sensory reliability trial by trial via gain variables g multiplying the mean responses of the input populations, with higher gains corresponding to more reliable sensory information (Figure 1a). We later consider alternative ways of manipulating the sensory reliability (see “Alternative representations of sensory reliability” below).”

- You end up concluding that your network works regardless of how reliability is encoded but that appears too strong since you haven't considered qualitatively different schemes, like different neurons representing parameters like stimulus mean and variance, or neural sampling based codes.

Yes, this claim has now been qualified in the revision (page 6): “These results provide evidence for the robustness of our approach to variations in the way in which sensory reliability is encoded in the input populations.” Although we test for three different ways of representing sensory reliability, we cannot exclude that our networks won't work with other schemes such as the ones the reviewer mentions.

- Also, how does this generalize to multidimensional inputs?

The extension to multi-dimensional stimuli is essentially trivial: the tuning of the input neurons would need to be multi-dimensional, but nothing else about the networks needs to be changed to handle multi-dimensional stimuli.

- non-prob reference model not explained at all in Results

We had, in fact, described the non-probabilistic reference models in the Results section of the old version: “To make sure that optimal performance in our tasks cannot be easily mimicked by heuristic, non-probabilistic models, we also calculated the performance of non-probabilistic reference models that did not take the reliabilities of the inputs into account (Figure 2c-d, blue). The large performance gaps between these non-probabilistic models and the optimal model suggest that approaching optimal performance in our tasks requires performing truly probabilistic inference, i.e. taking the reliabilities of the inputs into account.”

In the revision, we added more details about the non-probabilistic reference models to this paragraph. The paragraph now reads as follows (page 4): “To make sure that optimal performance in our tasks cannot be easily mimicked by heuristic, non-probabilistic models, we also calculated the performance of non-probabilistic reference models that did not take the reliabilities of the inputs into account (Figure 2c-d, blue). In continuous tasks, the non-probabilistic models estimated the individual cues or inputs optimally, but combined them sub-optimally by weighing them equally regardless of their reliability. Note that these models still performed a non-trivial probabilistic computation, namely marginalizing out a nuisance variable, i.e. the input gain, to come up with the optimal estimate of the individual cues. Similarly, in categorical tasks, the non-probabilistic models replaced the different reliability terms for different inputs in the optimal decision rules by a common reliability term (*Methods*). The large performance gaps between these non-probabilistic models and the optimal model suggest that

approaching optimal performance in our tasks requires correctly taking the reliabilities of the inputs into account.”

- Fig 2a: units? why different ranges on x axis?
- assumes these tasks are totally generic
- gain ranges explained too late, needed here!
- what g-ratio used for 2a results? ratio=0, and =1 of particular interest

Figure 2 is explained better in the revision. Gain ratios included a wide range of ratios from 5 to 0.2 in Figure 2a.

- Fig 4a and main text: "Note that we use relatively high gains in this particular example to make the optimal combination rule approximately linear. In the low gain conditions used in the main simulations, the optimal combination rule is no longer linear." This seems to be an important point that I don't understand. Please unpack.

For high gains (high reliability), the likelihoods are approximately Gaussian and the well-known linear combination rule (according to the reliabilities of the cues) applies. However, for low gains (the scenario adopted in the main simulations in the paper), the likelihoods are no longer Gaussian (due to edge effects) and the optimal combination rule is no longer linear.

- "We first note that in tasks with continuous output variables, the optimal solution is invariant to a multiplicative scaling g of the input responses (Methods, Equations 5-8)."
-> You state it as if it were a law of nature, whereas it's simply how you have set up your input representations.

Yes, it should be clear from the context that this refers to our main experiments with purely multiplicative gain in the input populations. In the revision, it is restated to make this explicit (page 6): "This discussion applies only to our main experiments where sensory reliability in the input populations is manipulated through purely multiplicative gain."

"We finally note that these results are decoder dependent. In the continuous tasks, for example, if we use a divisively normalized decoder instead of a linear read-out, posterior uncertainty is encoded in the mean hidden layer response, rather than in the sparsity of hidden layer responses (supplementary Figure S4)." -> Doesn't this imply that little meaning can be attached to this analysis since we simply don't know what the appropriate decoder is?

As stated in the paper, the sparsity-based representation of posterior uncertainty is conditional on a linear read-out. The assumption of a linear read-out is quite commonly used in the literature (e.g. Jazayeri & Movshon, *Nat Neurosci* 2006; DiCarlo and Cox, *TICS* 2007; Graf et al., *Nat Neurosci* 2011; Berens et al., *J Neurosci* 2012; Haefner et al., *Nat Neurosci* 2013; Pitkow et al., *Neuron* 2015), hence is not wholly unrealistic. Appropriate references are added on page 8 of the revised manuscript.

- Section on recurrent EI-networks potentially very interesting but totally insufficiently explained. Neither in the Results, nor in the Methods does it even say whether these are spiking neurons (LIF?), or binary units. Raises many more questions than it answers and probably best left for another paper.

In these networks, the neurons are rate neurons. The set-up is essentially the same as in the Song, Yang & Wang (2016) PLoS Comp. Biol. paper cited in this section. We think that the results in this section are important to establish the generality and robustness of our results: for example, lack of biological plausibility in generic feedforward networks is a recurring complaint we hear from our experimentalist colleagues. Therefore, we are against removing this section from the manuscript. In the revision, we provide more details about the simulations in the Methods section (page 18): "In the recurrent EI networks, all connections were constrained to satisfy Dale's law, as described in Song, et al. (2016). The recurrent units are all rate-based neurons. The ratio of excitatory to inhibitory neurons in both input and recurrent populations was 4 to 1. Inputs were presented over 10 time steps, but the total input information was equated to the total input information in the feedforward networks. The network's estimate was obtained from its output at the final, i.e. 10th time step. Other details are the same as in the corresponding simulations in the feedforward case."

- Discussion claims: "Our categorical networks already satisfy this principle, and our continuous networks satisfy it if, instead of a purely linear decoder, we use a linear decoder that is then normalized by the total activity in the hidden layer (supplementary Figure S4)"
-> Where is it shown that intermediate and output layers follow Poisson statistics?

We should first clarify that by Poisson-like variability, we mean exponential-family variability with linear sufficient statistics, i.e. sufficient statistics of the posterior can be linearly decoded from the neural responses (this is now explained in the Discussion). In supplementary figure 4, it is clear that the mean hidden layer responses and the mean input responses are approximately linearly related. Given that the inputs are Poisson distributed with linear sufficient statistics, the sufficient statistics of the posterior must also be approximately linearly decodable from the hidden layer responses. This is explained more clearly in the figure caption in the revision.

- 3rd empirical prediction: how can the dependence depend on the task? Let's say the animal has learnt both tasks and either switches between them or performs them concurrently. How could the responses be task-dependent?

We don't see a contradiction here. A separate population of neurons that provide information about the task context could switch the hidden layer responses from one type (say, mean activity-based encoding of uncertainty) to the other type (sparsity-based encoding of uncertainty). A similar context-dependent response switching mechanism was proposed by E. Salinas (*J Neurosci*, 2004) before.

Reviewer 3

I appreciate the effort that the authors have put into revising their paper, and the revised version is an improvement over the previous version of the manuscript. The more detailed elucidation of how these results go beyond previous work, the addition of simulations using other loss functions in the continuous case, and the inclusion of more biologically realistic neural network models are all things that strengthen the paper. However, for me the two issues that motivated my original review remain: this paper could either be a theoretical paper showing the properties of abstract neural network models, or a neuroscience paper connecting ideas about these models to how the brain works. I still feel like the contributions fall short of the mark under either of these construals.

As noted in the paper, the key theoretical contribution is the demonstration that the correspondence with posteriors is robust across training methods, generalization, and different paradigms. However, I still feel that this is a relatively incremental contribution that wouldn't come as a surprise to most researchers working with neural networks. As the authors note, similar robustness results were already known to hold for the discrete case.

This leaves things to turn on the contributions to neuroscience, and despite the addition of the new simulations these connections are speculative at best. I continue to feel that there's a nice paper here, but one that turns on deepening the connections to neuroscience rather than emphasizing the theoretical results.

The reviewer is making a subjective significance judgment about the contributions of our paper. We respectfully disagree with the reviewer and summarize here what we consider to be several novel and significant contributions made by our paper:

- We show that an error-based learning can account for the learning curve and qualitative aspects of the choice behavior of an animal learning a probabilistic inference task. To our knowledge, this hasn't been experimentally shown before and it's not a trivial result, as error-based learning has sometimes been argued to be too slow to be realistic (it is certainly too slow to be a realistic model of how humans learn to do similar tasks).
- We show that generic neural networks are much more efficient (in terms of the network size required for a given level of performance) than hand-crafted networks proposed in the literature before (including in some high-profile papers like Ma, Beck et al., Nat Neurosci 2006).
- We make an honest and serious attempt to understand how the networks solve probabilistic inference tasks, rather than accepting the solutions as black boxes, as is typically done in more practical deep learning applications. Along the way, we discover a novel sparsity-based population code for posterior uncertainty.
- As the reviewer notes, through various controls, we show that our results are robust and general: more specifically we consider alternative ways of representing uncertainty in the input populations, we consider variations on the architecture: random feedforward networks and recurrent EI networks, we consider different loss functions.

- More generally, we show that achieving near-optimal probabilistic inference might be much easier than people had thought before: a simple error-based learning scheme in small generic networks naturally leads to near-optimal probabilistic inference. We think that this could be an important insight in itself for the experimental neuroscientists and psychologists.

The reviewer suggests that we should try to deepen the paper's connections to neuroscience. Because the reviewer doesn't make any specific suggestions, we don't quite know how to do this. If the reviewer means that we should try to make more specific experimental predictions than the ones discussed in the Discussion section of the paper, for example, we see two problems with that: first, this would almost certainly mean that we need to constrain our networks with ever more biological details (different cell types, their physiological properties and connectivity patterns etc.) and this would go against our genericity claim (and introduce additional assumptions); second, even if one incorporates all these additional constraints (many of them still poorly understood, we might add), we suspect that training such networks on a single task or a small set of similar tasks would lead to a large number of degenerate solutions (the brain did not evolve to do just this specific set of tasks), hence would not lead to many meaningful general principles other than the ones we have already discovered with our generic networks, for example the sparsity-based representation of uncertainty in a subset of our tasks, and would instead get us bogged down in irrelevant details.

Reviewers' Comments:

The Reviewer did not have any comments to the authors.